# Assessing the contribution of rare protein-coding germline variants to prostate cancer risk and severity in 37,184 cases

Jonathan Mitchell [1,21] ✉, Niedzica Camacho [1,21], Patrick Shea[2], Konrad H. Stopsack [3,4], Vijai Joseph [5,6,7], Oliver S. Burren[1], Ryan S. Dhindsa [1,8], Abhishek Nag[1], Jacob E. Berchuck [9], Amanda O'Neill[1], Ali Abbasi[1], Anthony W. Zoghbi[8], Jesus Alegre-Díaz [10], Pablo Kuri-Morales [10,11], Jaime Berumen [10], Roberto Tapia-Conyer [10], Jonathan Emberson [12], Jason M. Torres [12], Rory Collins[12], Quanli Wang[13], David Goldstein [2], Athena Matakidou[1], Carolina Haefliger [1], Lauren Anderson-Dring[1], Ruth March[14], Vaidehi Jobanputra[2,15], Brian Dougherty[16], Keren Carss [1], Slavé Petrovski [1], Philip W. Kantoff[6,17,22], Kenneth Offit[5,6,7], Lorelei A. Mucci[4,18,22], Mark Pomerantz[9,22] & Margarete A. Fabre [1,19,20] ✉

To assess the contribution of rare coding germline genetic variants to prostate cancer risk and severity, we perform here a meta-analysis of 37,184 prostate cancer cases and 331,329 male controls from five cohorts with germline whole exome or genome sequencing data, and one cohort with imputed array data. At the gene level, our case-control collapsing analysis confirms associations between rare damaging variants in four genes and increased prostate cancer risk: *SAMHD1*, *BRCA2* and *ATM* at the study-wide significance level ($P < 1×10^{-8}$), and *CHEK2* at the suggestive threshold ($P < 2.6×10^{-6}$). Our case-only analysis, reveals that rare damaging variants in *AOX1* are associated with more aggressive disease (OR = 2.60 [1.75–3.83], $P = 1.35×10^{-6}$), as well as confirming the role of *BRCA2* in determining disease severity. At the single-variant level, our study reveals that a rare missense variant in *TERT* is associated with substantially reduced prostate cancer risk (OR = 0.13 [0.07–0.25], $P = 4.67×10^{-10}$), and confirms rare non-synonymous variants in a further three genes associated with reduced risk (*ANO7*, *SPDL1*, *AR*) and in three with increased risk (*HOXB13*, *CHEK2*, *BIK*). Altogether, this work provides deeper insights into the genetic architecture and biological basis of prostate cancer risk and severity, with potential implications for clinical risk prediction and therapeutic strategies.

Prostate cancer is the second most common cancer in men globally, with over 1.5 million new cases and 397,000 deaths estimated in 2022[1,2]. Whilst the majority of men diagnosed with localised disease are either cured or survive their cancer for many years, the 5-year survival in metastatic cases is just 30% and a substantial number live with treatment-related morbidity[3,4].

The pathogenesis of prostate cancer involves complex interactions between inherited genetic features, acquired somatic mutations and environmental factors. An important role for the germline genome is evident by the high heritability of prostate cancer risk, estimated by twin studies at 57%[5]. While genome-wide association studies (GWAS) have identified 451 variants to date, a large proportion of the

heritability remains unaccounted for[6–8]. Rare protein-coding germline variants associated with disease, compared to common variants, have larger effect sizes and often directly implicate causal genes[9], making rare variant disease associations particularly valuable for understanding mechanism and, as a result, identifying drug targets and elucidating treatment response[10,11]. For prostate cancer, linkage and candidate gene studies have identified influential rare variants in a small number of specific genes, such as *HOXB13* and *BRCA2*[12,13]. Importantly, emerging evidence suggests that the set of genes influencing the risk of developing prostate cancer is, at least in part, distinct from genes influencing prostate cancer aggressiveness[8,14]. For example, a genetic risk score incorporating disease risk variants was not associated with severity in men of European, Asian and Hispanic ancestries and did so only modestly in men of African ancestry, suggesting that additional genetic variants, not captured by the genetic score for risk of disease development, might influence disease behaviour[8].

In this work, to assess the contribution of rare germline variants exome-wide to the development of prostate cancer and its severity, we first test for rare variant associations at the gene level, utilising global biobanks, curated disease cohorts and clinical trial participants with germline whole exome sequencing (WES) or whole genome sequencing (WGS) data (total 19,926 cases; 187,705 controls). Subsequently, we incorporate imputed array data from the FinnGen cohort[15], a population enriched in low-frequency deleterious variants, to test for association at the single variant level (total 33,608 cases; 309,439 controls). Our study represents, to the best of our knowledge, the most comprehensive assessment to date of the role rare coding germline variants play in prostate cancer pathogenesis, and allows us to confirm previously reported genes associated with prostate cancer risk and severity, and implicate a role for genes not previously reported.

## Results
### Gene-level association testing
To investigate the aggregated influence of rare germline variants on prostate cancer risk and severity at the level of individual genes, we meta-analysed WES and WGS data from five cohorts totalling 19,926 prostate cancer cases and 187,705 male controls that met all quality control criteria (Table 1, see 'Methods'). These cohorts comprised The UK Biobank (UKB)[16,17], The Mexico City Prospective Study (MCPS)[18,19], The 100,000 Genomes Project (100 kGP)[20,21], three cohorts within the New York-Boston-AstraZeneca (NYBAZ) prostate cancer study, and a collection of AstraZeneca clinical trial (AZCT) participants. Except MCPS, which predominantly comprises individuals with Admixed American ancestry, the cohorts are primarily of European ancestry. However, we additionally included African, East Asian and South Asian strata where sufficient numbers of individuals were available.

Gene-phenotype association testing was performed under the previously described collapsing analysis framework[22,23]. To maximise discovery across potential genetic architectures, we included eleven qualifying variant (QV) models for each gene (ten dominant and one recessive), which filtered variants on a range of predicted effects and population frequency thresholds (Supplementary Data 1). The threshold for a suggestive association was set at $P < 2.6 \times 10^{-6}$ (corresponding to an exome-wide Bonferroni correction, 0.05/18,948 genes), and the study-wide significant threshold at the more stringent $P < 1 \times 10^{-8}$, which we have previously shown to result in an extremely low false positive rate when testing multiple QV models across multiple traits[23].

We first tested for genes associated with the overall risk of developing prostate cancer overall in a case-control analysis (19,926 cases vs 187,705 controls). The approach was robust, with no significant inflation in test statistics across the eleven QV models ($\lambda_{mean} = 1.04 \pm 0.024$, Supplementary Fig. 1 and Supplementary Data 2). We identified rare protein-truncating variants (PTVs) in the DNA

damage response (DDR) genes *BRCA2* (OR = 3.23 [2.65–3.90], $P = 7.5 \times 10^{-29}$) and *ATM* (OR = 2.92 [2.34–3.63], $P = 1.17 \times 10^{-19}$) and additionally rare damaging variants in *SAMHD1* (OR = 2.02 [1.65–2.45], $P = 2.36 \times 10^{-11}$) as significantly associated with increased prostate cancer risk (Figs. 1, 2, Table 2, Supplementary Fig. 2 and Supplementary Data 3–4). Rare damaging variants in *CHEK2* (OR = 1.69 [1.41–2.01], $P = 2.69 \times 10^{-8}$) and rare synonymous variants in *DMD* (OR = 0.50 [0.36–0.67], $P = 8.6 \times 10^{-7}$) were associated with prostate cancer risk at the suggestive significance threshold. *TET2* was also significantly associated with prostate cancer risk (OR = 3.31 [2.26–4.78], $P = 1.71 \times 10^{-9}$). However, the strong correlation between *TET2* carrier status and age (UKB EUR cohort: $P = 3.25 \times 10^{-5}$), and the skewed distribution of alternate reads percentage to below 50% (Supplementary Fig. 3), indicates a somatic mutational process. Indeed, while our analysis is confounded by age, the causal association of clonal somatic variants in the well-established clonal haematopoiesis (CH) driver gene *TET2* and prostate cancer has been described previously[24].

Consistent with the known importance of the DDR pathway in prostate cancer pathogenesis[13], we found *BRCA2*, *ATM* and *CHEK2* to be among the most significant risk genes. In the UKB cohort, 267/14,577 (1.8%) individuals who developed prostate cancer carried a QV in one of these three genes, compared to 900/115247 (0.8%) controls ($P_{FET} = 1.12 \times 10^{-29}$). We found no significant association with any additional DDR genes (Supplementary Fig. 4 and Supplementary Data 5). However, this could be due to a lack of power related to low carrier frequency; in *MSH2*, for example, it is notable that the effect size estimate for PTVs (OR = 3.38 [1.55–6.90], $P = 1.20 \times 10^{-3}$) was similar to the effect sizes in DDR genes found to be significantly associated with the overall risk of prostate cancer.

Next, using the available clinical data for the five cohorts, we stratified cases into aggressive prostate cancer (agg. PCa) and non-aggressive prostate cancer (non-agg. PCa), a distinction we refer to subsequently as 'severity'. Aggressive prostate cancer was defined if any one of a number of criteria were met: tumour stage T4 or N1 or M1, Gleason score ≥ 8, prostate cancer as primary cause of death, prostate cancer treated with chemotherapy, or castration-resistant prostate cancer (see 'Methods'). We performed a case-only gene-level association test across the exome (4207 agg. PCa cases vs 15,170 non-agg. PCa cases, Table 1), to identify genes associated with disease severity (Table 2, Supplementary Data 6–8 and Supplementary Figs. 5–7). PTVs in *BRCA2* were significantly associated with increased severity (OR = 3.82 [2.70–5.41], $P = 1.58 \times 10^{-14}$), as were rare damaging variants in *AOX1* at the suggestive level (OR = 2.60 [1.75–3.83], $P = 1.35 \times 10^{-6}$). Beyond *BRCA2*, of the genes found to be associated with prostate cancer risk, the DDR gene *ATM* showed the strongest evidence of also being associated with severity (OR = 2.23 [1.47–3.34], $P = 9.41 \times 10^{-5}$, Fig. 2, Supplementary Fig. 8 and Supplementary Data 9). Indeed, in UKB European cohort, we found that 2.9% (48/1641) of all aggressive prostate cancer cases carried a *BRCA2* or *ATM* QV compared to 0.9% (114/12,936) of non-aggressive prostate cancer cases ($P_{FET} = 1.46 \times 10^{-10}$).

Finally, at the gene-level, we tested for genetic association between aggressive prostate cancer and controls, and found that PTVs in *BRCA2* (OR = 8.23 [6.17–10.85], $P = 1.47 \times 10^{-36}$) and *ATM* (OR = 5.27 [3.65–7.46], $P = 1.74 \times 10^{-16}$) were significantly associated with aggressive disease (Table 2, Supplementary Data 10–13, Supplementary Figs. 9–12). Consistent with *BRCA2* and *ATM* showing association with disease severity, their effect sizes were larger in this aggressive prostate cancer versus controls analysis compared to the overall prostate cancer versus controls analysis (Fig. 2 and Table 2).

Leveraging variant type to infer direction of effect, our observation that the QV model most significantly associated with prostate cancer risk and severity in *BRCA2* and *ATM* was the 'ptv model' (containing only PTVs), suggests that these genes operate via a loss-of-function mechanism in prostate cancer (Table 2, Supplementary

**Table 1 | Sample size, ancestry and genetic data type of all cohorts used in the gene-level and single variant-level genetic association meta-analyses**

| Cohort | Genetic Ancestry | Genetic Data | Total PCa Cases (n) | Agg. PCa (n) | non-Agg. PCa (n) | Controls (n) | Variant-level testing inclusion |
|---|---|---|---|---|---|---|---|
| UKB | EUR | WES | 14,577 | 1641 | 12,936 | 115,247 | Yes |
| MCPS | AMR | WES | 282 | 181 | 101 | 35,801 | Yes |
| 100kGP | EUR | WGS | 1011 | 83 | 928 | 8759 | Yes |
| NYBAZ Study | EUR | WES | 2200 | 995 | 1205 | 17,600 (UKB) | No |
| AZCT | EUR | WES | 1230 | 1230 | 0[a] | 3226 | No |
| UKB | AFR | WES | 349 | - | - | 2119 | Yes |
| UKB | SAS | WES | 131 | - | - | 3889 | Yes |
| AZCT | EAS | WES | 77 | 77 | - | 650 | No |
| NYBAZ Study | AFR | WES | 69 | - | - | 414 (UKB) | No |
| Gene-level testing total | - | - | **19,926** | **4207** | **15,170** | **187,705** | - |
| FinnGen | Finnish | Imputed Genotypes | 17,258 | - | - | 143,624 | Yes |
| Variant-level testing total | - | - | **33,608** | **1905** | **13,965** | **309,439** | - |

Aggressive prostate cancer (Agg. PCa) is defined by tumour stage T4/N1/M1, Gleason score ≥ 8, prostate cancer as underlying cause of death, metastatic prostate cancer, prostate cancer treated with chemotherapy or castration resistant prostate cancer.

*PCa* prostate cancer, *non-Agg. PCa* non-aggressive prostate cancer, *UKB* UK Biobank, *MCPS* Mexico City prospective Study, *100kGP* 100,000 Genomes Project, *NYBAZ Study* New York-Boston-AstraZeneca prostate cancer study, *AZCT* AstraZeneca Clinical Trials, *EUR* European, *AMR* Admixed American, *AFR* African, *SAS* South Asian, *EAS* East Asian, *WES* whole exome sequencing, *WGS* whole genome sequencing.

[a]For the clinical trial cohort Agg. PCa Vs non-Agg. PCa analysis in EUR, a subset of non-aggressive UKB cases were used as this cohort contained none.

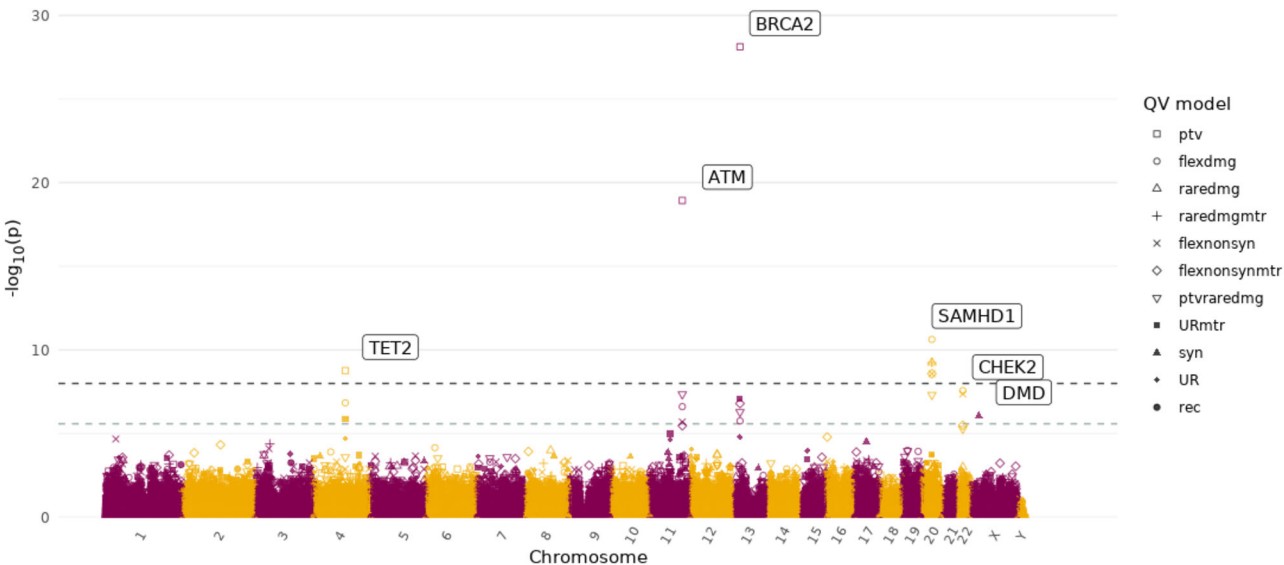

**Fig. 1 | Manhattan plot of all meta-analysis gene-level association tests with the risk of developing overall prostate cancer.** The x-axis is the genomic position of the gene and the y-axis is the -log$_{10}$ transformed unadjusted *P* values for all qualifying variant (QV) models (defined in Supplementary Data 1) as indicated in the legend. *P* values were determined from a Cochran–Mantel–Haenszel test across cohorts. The light grey dashed line represents the suggestive significance threshold ($P = 2.6 \times 10^{-6}$) and the dark grey dashed line the study-wide significance threshold ($P = 1 \times 10^{-8}$). Genes which reach the suggestive significance threshold are labelled, and only the most significant QV model for each gene is labelled. ptv = rare protein-truncating variant QV model; flexdmg = rare damaging non-synonymous QV model.

Data 1). For three additional genes—*CHEK2*, *SAMHD1* and *AOX1*—the most significant QV model included a combination of rare predicted damaging missense and PTVs ('flexdmg', Table 2, Supplementary Data 1). To assess whether these associations were also operating via a loss-of-function mechanism, we looked for evidence in the QV model that includes only PTVs ('ptv'), and observed associations at $P < 0.05$ in a consistent direction in all three cases (*CHEK2*, OR = 1.58 [1.00–2.41], $P = 0.035$; *SAMHD1*, OR = 2.15 [1.22–3.63], $P = 0.006$; *AOX1*, OR = 3.63 [1.36–9.63], $P = 0.006$, Supplementary Data 4 and 8). In all genes found to be associated with prostate cancer in our analyses, the most significantly associated QV model included a large number of separate QVs distributed along the amino acid sequence (Supplementary Figs. 13–14).

**Protein-coding variant-level association testing**

We next performed a variant-level, exome-wide association study (ExWAS) to identify individual rare variants associated with prostate cancer. We analysed sequencing data from the cohorts included in the gene-level analysis and, additionally, imputed genotype array data from the FinnGen cohort[15]. The meta-analysis was restricted to cohorts

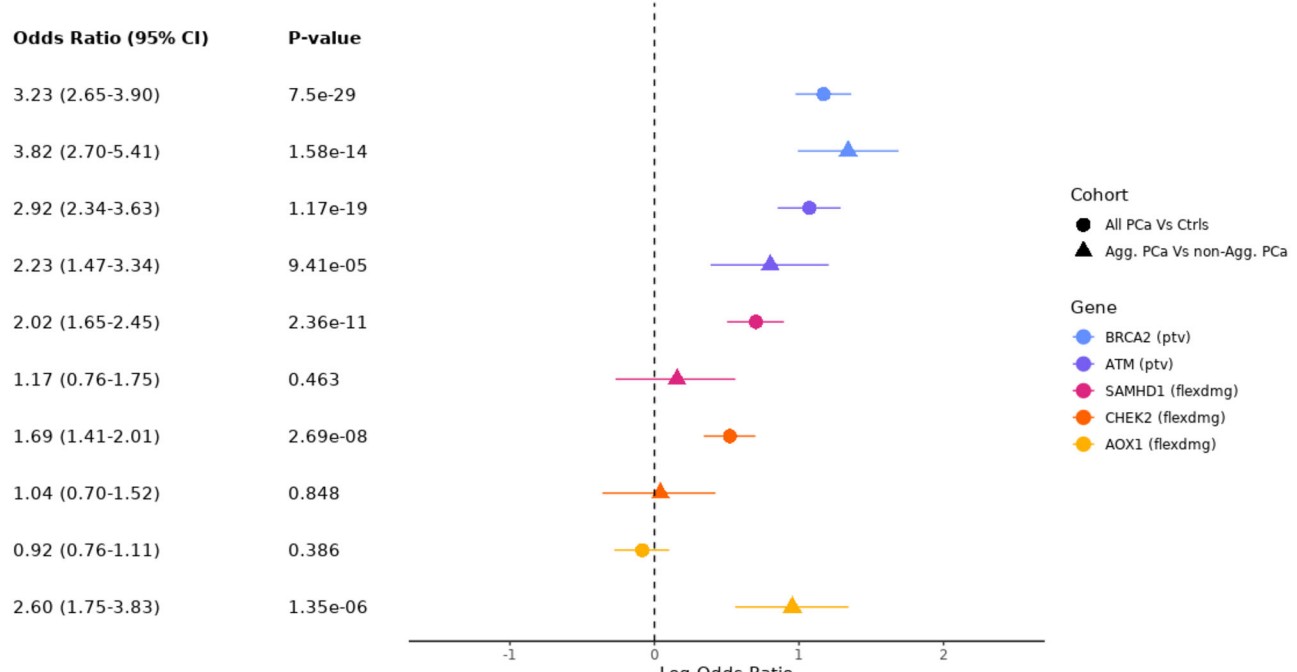

| Odds Ratio (95% CI) | P-value |
|---|---|
| 3.23 (2.65–3.90) | 7.5e-29 |
| 3.82 (2.70–5.41) | 1.58e-14 |
| 2.92 (2.34–3.63) | 1.17e-19 |
| 2.23 (1.47–3.34) | 9.41e-05 |
| 2.02 (1.65–2.45) | 2.36e-11 |
| 1.17 (0.76–1.75) | 0.463 |
| 1.69 (1.41–2.01) | 2.69e-08 |
| 1.04 (0.70–1.52) | 0.848 |
| 0.92 (0.76–1.11) | 0.386 |
| 2.60 (1.75–3.83) | 1.35e-06 |

**Fig. 2 | Forest plot showing the association of genes with prostate cancer risk (All PCa Vs Ctrls) and severity (Agg. PCa Vs non-Agg. PCa) which reached the suggestive significance threshold ($P = 2.6 \times 10^{-6}$) for non-synonymous qualifying variant (QV) models.** Odds ratios and $P$ values were determined from a Cochran–Mantel–Haenszel test across cohorts. Gene and QV model (defined in Supplementary Data 1) are as indicated in legend. For genes where more than one QV model passed the suggestive significance threshold the most significant is plotted. ptv = rare protein-truncating variant QV model; flexdmg = rare damaging non-synonymous QV model. PCa prostate cancer, Agg aggressive, Non-agg non-aggressive.

which did not have a high level of genomic inflation within the ExWAS ($\lambda < 1.15$, Supplementary Data 14, see 'Methods'), resulting in a total of 33,608 prostate cancer cases and 309,439 male controls. We tested 1,573,300 variants using three genetic models (additive, dominant and recessive, see 'Methods'), and set a threshold of $P < 1 \times 10^{-8}$ for study-wide statistical significance[23].

We identified 92 variants associated with the risk of developing prostate cancer at the study-wide significance threshold, of which sixteen were rare (minor allele frequency (MAF) < 1%) in non-Finnish Europeans (Fig. 3, Supplementary Data 15). These sixteen rare protein-coding variants were spread over eight loci, and there was statistical evidence for seven of them being the causal variant in the locus (FinnGen SuSiE[25] posterior inclusion probability (PIP) > 0.05, Table 3). One of the sixteen variants (17:47809406:G:A in *OSBPL7*) was not present in FinnGen, with the association being driven only by the UKB ExWAS ($P = 2.52 \times 10^{-11}$), and was not significant ($P = 0.12$) after conditioning on the lead variant (17:48728343:C:T in *HOXB13*) in the locus.

All seven putatively causal variants were non-synonymous: a frameshift variant in *CHEK2*, missense variants in *HOXB13*, *ANO7*, *SPDL1*, *AR* and *TERT*, and a conservative inframe deletion in *BIK* (Table 3, Supplementary Data 15). In FinnGen, all seven variants were significantly associated with prostate cancer risk ($P < 1 \times 10^{-8}$), and for four of the variants there was evidence of association ($P < 0.05$) after excluding FinnGen and meta-analysing the sequenced cohorts alone (Supplementary Data 15). Although the significantly associated *BIK* conservative inframe deletion variant was unique to the FinnGen cohort, a separate rare disruptive inframe deletion in the same gene was present in UKB (22:43129228:GTGCTGCTGGCGCTGCTGC:G, OR = 1.49 [1.20–1.85], $P = 6.20 \times 10^{-4}$). The variants in *HOXB13*, *ANO7*, *CHEK2*, *SPDL1*, *AR* and *BIK* have been previously reported[8,15], while the protective missense variant in *TERT* is novel (OR = 0.134 [0.071–0.252], $P = 4.67 \times 10^{-10}$).

For the sequenced cohorts, meta-analyses for the dominant and recessive models were performed, but did not reveal any additional statistically associated variants. In the case-only and case-control analyses of aggressive prostate cancer, which were limited to the UKB, MCPS and 100,000 Genomes Project cohorts, there were no significantly associated rare variants.

## Discussion
Our meta-analysis of 37,184 prostate cancer cases and 331,329 controls —derived from global biobanks, clinical trials and curated disease cohorts—represents, to the best of our knowledge, the most comprehensive assessment of the role of rare germline variants in prostate cancer risk and severity to date. While several DDR genes are established as conferring prostate cancer risk and are included in clinical guidelines for germline genetic testing[26], the significance of genes beyond *BRCA2* is not well understood[13]. Here, we validate *BRCA2*, *ATM* and *CHEK2* deleterious rare variants as significant risk factors, and reproduce the recently described association of *SAMHD1*[27] with prostate cancer in UKB and replicate the finding in additional cohorts. It is notable that the QV model strongly associating *SAMHD1* with prostate cancer risk here is the same model we recently found to be associated with longer telomere length[28]. Given the widely reported links between telomere biology and cancer[29–31], in particular the association between longer genetically predicted leucocyte telomere length and increased prostate cancer risk[29], telomere maintenance is implicated as a potential mechanism for *SAMHD1*-mediated predisposition to prostate cancer. At the gene level, we also identified *TET2* and *DMD* to be associated with the risk of prostate cancer. We demonstrated that the *TET2* association was due to somatic variants, and although the Duchenne muscular dystrophy (DMD) gene has been previously implicated in cancer[32], the association we report here is for synonymous variants and at the suggestive level and should therefore be interpreted with caution.

**Table 2 | Genes significantly associated at the suggestive significance level ($P < 2.6 \times 10^{-6}$) with risk of developing prostate cancer (PCa Vs Ctrls) and/or its severity (Agg. PCa Vs non-Agg. PCa)**

| Gene | QV Model | Association Analysis | P | OR [95% CI] | Case Carrier Frq. | Ctrl Carrier Frq. |
|---|---|---|---|---|---|---|
| BRCA2 | ptv | PCa Vs Ctrls | $7.50 \times 10^{-29}$ | 3.23 [2.65–3.90] | 0.00793 | 0.00248 |
| | | Agg. PCa Vs non-Agg. PCa | $1.58 \times 10^{-14}$ | 3.82 [2.70–5.41] | 0.0194 | 0.00492 |
| | | Agg. PCa Vs Ctrls | $1.47 \times 10^{-36}$ | 8.23 [6.17–10.85] | 0.0188 | 0.00246 |
| ATM | ptv | PCa Vs Ctrls | $1.17 \times 10^{-19}$ | 2.92 [2.34–3.63] | 0.00607 | 0.00225 |
| | | Agg. PCa Vs non-Agg. PCa | $9.41 \times 10^{-05}$ | 2.23 [1.47–3.34] | 0.0111 | 0.00478 |
| | | Agg. PCa Vs Ctrls | $1.74 \times 10^{-16}$ | 5.27 [3.65–7.46] | 0.0116 | 0.00221 |
| SAMHD1 | flexdmg | PCa Vs Ctrls | $2.36 \times 10^{-11}$ | 2.02 [1.65–2.45] | 0.00678 | 0.00322 |
| | | Agg. PCa Vs non-Agg. PCa | 0.463 | 1.17 [0.76–1.75] | 0.00823 | 0.00810 |
| | | Agg. PCa Vs Ctrls | $5.98 \times 10^{-3}$ | 1.80 [1.17–2.68] | 0.00737 | 0.00341 |
| TET2[a] | ptv | PCa Vs Ctrls | $1.71 \times 10^{-09}$ | 3.31 [2.26–4.78] | 0.00220 | $7.35 \times 10^{-4}$ |
| | | Agg. PCa Vs non-Agg. PCa | 0.836 | 0.812 [0.29–1.95] | 0.00169 | 0.00239 |
| | | Agg. PCa Vs Ctrls | 0.314 | 1.58 [0.57–3.69] | 0.00166 | $7.45 \times 10^{-4}$ |
| CHEK2 | flexdmg | PCa Vs Ctrls | $2.69 \times 10^{-8}$ | 1.69 [1.41–2.01] | 0.00793 | 0.00522 |
| | | Agg. PCa Vs non-Agg. PCa | 0.848 | 1.04 [0.70–1.52] | 0.00944 | 0.00904 |
| | | Agg. PCa Vs Ctrls | $8.04 \times 10^{-3}$ | 1.65 [1.13–2.34] | 0.00951 | 0.00510 |
| DMD | syn | PCa Vs Ctrls | $8.60 \times 10^{-07}$ | 0.50 [0.36–0.67] | 0.00226 | 0.00464 |
| | | Agg. PCa Vs non-Agg. PCa | 0.718 | 1.140 [0.52–2.33] | 0.00291 | 0.00239 |
| | | Agg. PCa Vs Ctrls | 0.0303 | 0.512 [0.25–0.95] | 0.00261 | 0.00449 |
| AOX1 | flexdmg | PCa Vs Ctrls | 0.386 | 0.92 [0.76–1.11] | 0.00642 | 0.00889 |
| | | Agg. PCa Vs non-Agg. PCa | $1.35 \times 10^{-6}$ | 2.60 [1.75–3.83] | 0.0128 | 0.00485 |
| | | Agg. PCa Vs Ctrls | $6.56 \times 10^{-3}$ | 1.54 [1.12–2.09] | 0.0121 | 0.00870 |

The qualifying variant (QV) model (defined in Supplementary Data 1) with the strongest association is shown. Carrier frequency is the fraction of individuals with at least one qualifying allele in the gene. ptv = rare protein-truncating variant QV model; flexdmg = rare damaging non-synonymous QV model; syn = rare synonymous QV model. Odds ratios and P-values were determined from a Cochran–Mantel–Haenszel test across cohorts.

*P* P-value from gene-level association test, *OR* odds ratio from gene-level association test, *95% CI* 95% confidence intervals, *PCa* prostate cancer, *Agg. PCa* aggressive prostate cancer, *non-Agg. PCa* non-aggressive prostate cancer.

[a]Association driven by somatic variants.

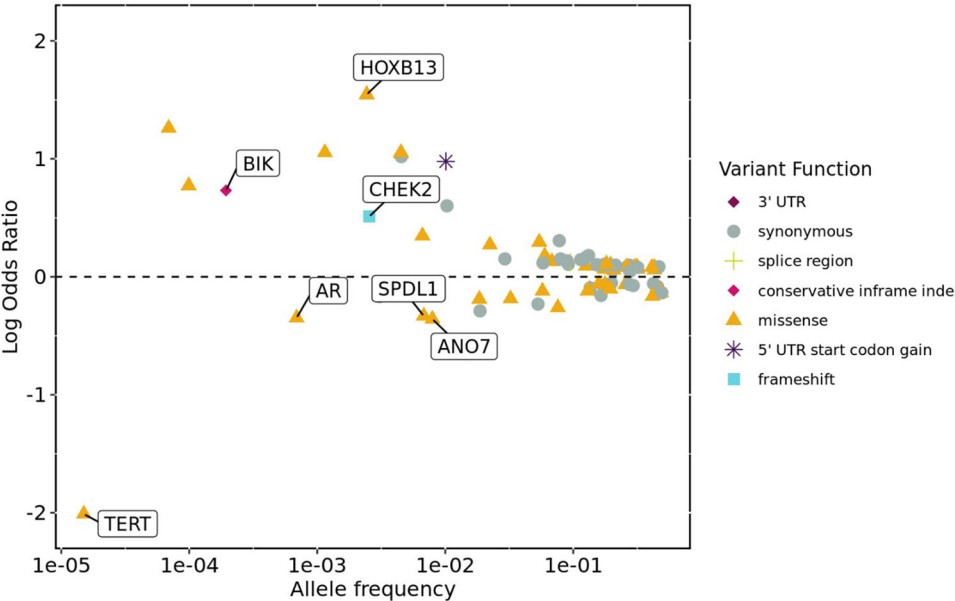

**Fig. 3 | Summary of exome wide association study variants which reached study-wide significance ($P < 1 \times 10^{-8}$) in the meta-analysis for the risk of developing prostate cancer.** The x-axis is the variant MAF in non-Finnish Europeans, and the y-axis is the variant effect estimate. Gene labelled variants are those which are rare in non-Finnish Europeans (MAF < 1%) and had a posterior inclusion probability of being a causal variant greater than 0.05 in the FinnGen study. The *P* value used to determine significance is from the Stouffer's meta-analysis and as this does not generate an effect-size we report here the effect estimate from the FinnGen cohort as calculated with REGENIE using Firth's logistic regression. MAF minor allele frequency, UTR untranslated region.

**Table 3 | Putatively causal rare (PIP > 0.05 and non-Finnish EUR MAF < 0.01) variants significantly associated at the study-wide level ($P < 1 \times 10^{-8}$) with the risk of prostate cancer**

| Gene | Protein Change | Variant (Chr:Pos:ref:alt, HGVS) | P | OR [95% CI] | MAF (non-Finnish EUR) | MAF (Finnish EUR) | PIP |
|---|---|---|---|---|---|---|---|
| HOXB13 | ENST00000290295:p.Gly84Glu | 17:48728343:C:T, NC_000017.11:g.48728343C>T | $1.95 \times 10^{-181}$ | 4.69 [4.22–5.21] | 0.00244 | 0.00786 | 1 |
| ANO7 | ENST00000274979:p.Glu226Lys | 2:241200185:G:A, NC_000002.12:g.241200185G>C | $4.57 \times 10^{-26}$ | 0.699 [0.659–0.741] | 0.00793 | 0.0619 | 0.985 |
| CHEK2 | ENST00000328354:p.Thr367fs | 22:28695868:AG:A, NC_000022.11:g.28695869del | $1.17 \times 10^{-20}$ | 1.67 [1.46–1.91] | 0.00255 | 0.00861 | 0.078 |
| SPDL1 | ENST00000265295:p.Arg20Gln | 5:169588475:G:A, NC_000005.10:g.169588475G>A | $3.06 \times 10^{-13}$ | 0.718 [0.663–0.777] | 0.00679 | 0.0346 | 0.996 |
| AR | ENST00000374690:p.Glu654Lys | X:6771476:G:A, NC_000023.11:g.6771476G>A | $1.28 \times 10^{-11}$ | 0.706 [0.655–0.761] | $6.90 \times 10^{-4}$ | 0.0194 | 0.604 |
| BIK | ENST00000216115:p.Ala139_Leu148del | 22:43129228: GTGCTGCTGGCGCTGCTGCTGCTGCTGCTGGCGC:G, NC_000022.11:g. 43129228del | $2.04 \times 10^{-11}$ | 2.08 [1.68–2.57] | $1.94 \times 10^{-4}$ | 0.00331 | 0.3 |
| TERT | ENST00000031058:p.Asp684Gly | 5:1279370:T:C, NC_000005.10:g.1279370T>C | $4.67 \times 10^{-10}$ | 0.134 [0.071–0.252] | $1.5 \times 10^{-5}$ | 0.00151 | 0.985 |

The P-value reported is from the Stouffer's meta-analysis and as this does not generate an effect-size we report here the effect estimate (OR) from the FinnGen cohort as calculated with REGENIE using Firth's logistic regression. Minor allele frequencies (MAF) from gnomAD non-Finnish European and Finnish-European populations.
Chr:Pos:ref:alt chromosome, genomics position (GRCh38), reference allele, alternate allele, HGVS human genome variation society nomenclature, OR odds ratio, 95% CI 95% confidence interval, PIP posterior inclusion probability as calculated by SuSiE in FinnGen.

At the single variant level, we identified a novel missense variant in *TERT* with a striking degree of protection (carriers have a 7.5-times lower odds compared to non-carriers of developing prostate cancer), providing further evidence that telomere maintenance plays a key role in prostate cancer development. In addition to *TERT*, we found rare non-synonymous variants in three genes associated with decreased prostate cancer risk (*ANO7*, *SPDL1* and *AR*), and three genes associated with increased risk (*HOXB13*, *CHEK2* and *BIK*). *ANO7* is a prostate-specific gene, and consistent with the protective *ANO7* missense variant reported here, an *ANO7* eQTL (2:241195850:G:A) common in the European population (MAF = 2.10%) has previously been found to be associated with both prostate cancer risk and severity[33]. *SPDL1* is involved in mitotic checkpoint signalling during cell division[34], and the *SPDL1* missense variant (5:169588475:G:A) protective for prostate cancer in our analysis has previously been shown to increase the risk of idiopathic pulmonary fibrosis (IPF)[35], consistent with existing literature on shared genetic alterations between cancer and IPF[36]. Finally, the protective missense variant in *AR*, which encodes the androgen receptor, is notable given the widespread treatment of prostate cancer patients with anti-androgen therapies[37], and highlights the connection between rare germline variant disease associations and potential therapeutic targets.

Analysing associations between germline variation and disease end-points provides insight into the distinct pathogenic roles of individual genes[14]. Specifically, we identify germline variants in the case versus control analysis that play a role in the overall risk of developing prostate cancer, while genetic variants identified in the within-case aggressive versus non-aggressive analysis play a role in prostate cancer severity. In our study, *BRCA2* was the only gene with clear evidence for a role both in the overall risk of prostate cancer and also in determining the degree of aggressiveness, consistent with previous reports[38]. In comparison, two other genes – *SAMHD1* and *CHEK2* – showed significant associations in the case-control analysis of prostate cancer risk but demonstrated no association with disease severity, similar to the reported effect of *HOXB13* p.Gly84Glu[39]. Conversely, damaging germline variants in *AOX1* were not associated with the overall risk of developing prostate cancer, but were associated with aggressive disease at the suggestive level. This is consistent with a prior GWAS identifying a common variant at the *AOX1* locus, which was associated with prostate-cancer-specific survival time, and with *AOX1* expression levels that correlated with disease recurrence[40]. If validated, this implies a role for *AOX1* in prostate cancer progression, but no substantial impact on the overall risk of disease development.

Our study has a number of potential limitations. Firstly, the gene-level association meta-analysis includes studies where the cases and the controls were recruited from separate cohorts. While this is a necessary approach for including disease specific cohorts in rare variant association studies, biases may be introduced from technical artefacts and population differences[41,42]. In this study, we mitigated these potential biases by using the same bioinformatics pipeline for cases and controls, and by using strict quality control criteria aimed at ensuring cohort harmonisation. Although it is not possible to entirely rule out that some bias remained, we did not observe significant genomic inflation in our association test statistics, and reassuringly there was evidence from multiple cohorts for all statistically significant associations. Secondly, in our single variant analysis, while the Finnish population represents a powerful bottleneck population for discovering low frequency disease-associated variants[15], the extreme rarity of many of these variants in non-Finnish European populations makes replication of findings challenging, even in large cohorts such as UKB. Thirdly, FinnGen's genotyping data is imputed and, although the imputation utilised a population-specific reference panel of high-coverage WGS data[15] and we excluded low quality imputed variants, findings derived from imputed variants should be interpreted with greater caution than those derived from direct sequencing. Finally, the

theoretical misclassification of (i) controls (which might have included individuals unknowingly destined to develop prostate cancer in future) and (ii) non-aggressive cases (which might have included individuals who would have developed features of aggressive disease had they not received treatment), potentially reduced our power to detect genetic signals.

Our findings have potential clinical implications that warrant further study. Inheritance of variants associated with prostate cancer risk, for example, could influence prostate cancer screening recommendations, with carriers potentially benefiting from earlier and/or more intensive testing. Similarly, inheritance of variants associated with aggressive prostate cancer could impact intensity of monitoring and/or treatment decisions. Both of these projections require further investigation in dedicated studies. Furthermore, the identification of pathogenic variants in specific genes/pathways could inform precision medicine strategies. Finally, clinical risk stratification tools will likely be improved by integration of rare germline variants identified here with previously established risk factors, including common germline variants, somatic tumour driver mutations and non-genetic patient features.

Overall, our analysis provides insights into the contribution of rare deleterious variants to prostate cancer risk and severity and, through the associated genes, into pathogenic mechanisms.

## Methods

### Cohorts

The research presented here complies with the ethical regulations approved for each cohort. The UKB has approval from the North-West Multi-centre Research Ethics Committee (11/NW/0382), and participants provided written informed consent[16]. The MCPS was approved by the Mexican National Council for Science and Technology, the Mexican Ministry of Health and the University of Oxford ethics committees, and participants provided written informed consent[18,19]. The 100,000 Genomes Project was approved by the National Research Ethics Committee, and participants provided written informed consent[21]. FinnGen study approval was obtained by the Coordinating Ethics Committee of the Hospital District of Helsinki and Uusimaa (number HUS/990/2017), and all participants provided informed consent[15]. All participants in the NYBAZ cohort provided informed consent[43]. All participants in the AZCT cohort provided written informed consent for DNA sequencing and the use of this data for research purposes.

We brought together data from prospective cohort studies of cancer-free men as well as clinical and epidemiologic studies of patients with prostate cancer. UKB is a prospective study which recruited ~500,000 participants between the ages of 40 and 65 years in the United Kingdom from 2006 until 2010, of whom 46% were male[16]. Each participant provided blood and urine samples. Additionally, data for each patient includes periodically updated electronic health records, health questionnaire results, and linkage to death and cancer registries.

MCPS is a cohort of ~150,000 participants recruited at 35 years of age or older in Mexico City from 1998 to 2004, of whom 33% are male[19]. Participants provided a blood sample, completed a health questionnaire and the study provided access to their death registry data (updated 2020).

The NYBAZ prostate cancer study consists of prostate cancer patients from three separate cohorts: participants of the Health Professionals Follow-up Study (HPFS) and the Physicians' Health Study (PHS) who were diagnosed with prostate cancer during prospective follow-up and patients with cancer seen at the Dana-Farber Cancer Institute (DFCI) Gelb Center. From these three studies, 2607 participants with high-risk prostate cancer were selected who had blood samples available.

HPFS and PHS are prospective cohorts that enroled men from across the US with a professional background in health professions (HPFS) and medicine (PHS). HPFS started in 1986 with 51,529 initially cancer-free men, collected blood samples in 1993–95 from 18,000 and continues to follow participants for cancer incidence and mortality. PHS started as randomised-controlled trials of aspirin and multivitamins in chronic disease prevention among 22,071 initially cancer-free men in 1982, with blood samples at baseline. Follow-up for both cohorts is similar, and prostate cancer diagnoses were confirmed by a review of medical records and pathology reports[44]. Causes of death were assigned by a physician endpoint committee based on medical records, reports from next-of-kin and the National Death Index. Data for this study included those with a prostate cancer diagnosis (1982–2014) with an available blood sample, who were high-risk (Gleason score ≥ 4 + 3 (grade groups 3–5), stage ≥ T3, or PSA ≥ 20 ng/ml), but no regional or distant metastases at diagnosis (cN0/Nx M0/Mx or pN0/Nx M0/Mx).

DFCI GELB is an observational clinical study that includes patients with prostate cancer seen in the medical oncology department since 1997. Demographic and clinical data were captured in a structured database by treating clinicians at enrolment or by research assistants from the electronic medical record during follow-up[45], with death follow-up via the National Death Index. Patients (1997–2018) were selected for sequencing if they had localised (N0 M0) prostate cancer at initial diagnosis, had undergone surgery or radiation, had at least one high-risk feature as in HPFS/PHS (except Gleason scores ≥ 8/grade group 4–5), had any repeat contact with DFCI (95%) and had survived ≥ 3 years after initial diagnosis.

The 100,000 Genomes Project recruited patients from the United Kingdom's National Health Service based on rare disease and cancer diagnoses[20,21]. Blood samples and clinical data were collected, and with consent participants were linked to electronic health records and the UK cancer registry.

The AZCT cohort contained a total of 1445 prostate cancer patients enroled across nine clinical trials: EPOC (NCT00090363), ENTHUSE M1 (NCT00554229), ENTHUSE M0 (NCT00626548), ENTHUSE M1C (NCT00617669), UVA97934; Study 8 (NCT01972217), PROpel (NCT03732820), MAD (NCT04087174), NCT04089553, AARDVARC (NCT04495179). All enroled patients were diagnosed with either metastatic prostate cancer, castration-resistant prostate cancer, or metastatic castration-resistant prostate cancer.

FinnGen is a research project encompassing 9 Finnish biobanks, and the results presented here are from ~445,000 participants included in FinnGen release 11[15]. Blood samples were collected from each participant and data from the Finnish nationwide longitudinal health register is available.

### Phenotypes

In the UKB, prostate cancer cases were identified from the cancer register (UKB Data-Field 40006), death register (UKB Data-Fields 40001 and 40002) and hospital inpatient diagnoses (UKB Data-Fields 41270) using International Classification of Diseases (ICD)-10 code C61, and additional cases from primary care records (Read v2). In the MCPS cohort cases were identified as participants self-reporting as diagnosed with prostate cancer in the baseline recruitment questionnaire and from the death register (ICD-10 code C61). In the 100,000 Genomes Project cohort cases were identified from those recruited to the project based on a diagnosis of prostate cancer. Additional cases were identified across the entire project cohort from linkage to the hospital episode statistics, the cancer register and the death register using ICD-10 code C61. All individuals in the AZCT cohort were recruited to the trials based on a diagnosis of prostate cancer. Finally, in the FinnGen cohort cases were identified from hospital discharge records, cause of death records and cancer registry using ICD-10 code C61 and ICD-9 code 185.

Controls in UKB were used for the UKB and NYBAZ cohorts of cases. These were defined as male participants without malignant neoplasm diagnoses, as defined by ICD-10 codes C00-C90 in the cancer register, hospital admissions and death register. Additionally, individuals were removed from the control set based on self-reported prostate cancer or family history of prostate cancer (father or brother diagnosed with prostate cancer). UKB controls for the NYBAZ cohort were selected based on those samples in UKB which best matched the total number of rare deleterious variants across the exome ('flexdmg' QV model as in Supplementary Data 1). In MCPS, controls were defined as male participants without prostate cancer. Controls for the AZCTs prostate cancer case cohort were comprised of male participants from non-oncology clinical trials in the cardiovascular, renal, metabolism, respiratory and immunology therapy areas. For the 100,000 Genomes Project, a set of controls was identified from the rare disease arm of the project. From these, male individuals who were not the proband and who had no prostate cancer diagnosis were selected. In the FinnGen study, male participants with no diagnoses of any cancer were used as controls.

In the UKB, MCPS, NYBAZ, 100,000 Genomes Project and AZ clinical trials cohorts, cases were stratified into non-aggressive prostate cancer and aggressive prostate cancer based on the available clinical data. In UKB and 100,000 Genomes Project, aggressive prostate cancer cases were defined as those with prostate cancer as the underlying cause of death or prostate cancer as the only primary neoplasm and a secondary neoplasm (ICD-10 codes C77, C78, C79) or prostate cancer and chemotherapy (based on OPCS Classification of Interventions and Procedures). In MCPS, aggressive prostate cancer was identified as those with prostate cancer as their underlying cause of death. In the NYBAZ cohort, individuals with tumour stage T4/N1 or Gleason score ≥ 8 were defined as aggressive prostate cancer. All participants in the AZCTs cohort were metastatic and/or castration-resistant and were therefore classified as aggressive prostate cancer cases.

### Sequencing, variant calling, genotyping and imputation

For all WES studies sequencing was performed using the IDT xGen v1 capture kit on the NovaSeq6000 platform. Both the UKB and MCPS cohorts were whole exome sequenced at the Regeneron Genetics Center with 75-bp paired ends[18,23,46]. The New York-Boston-AstraZeneca (NYBAZ) prostate cancer study samples were whole exome sequenced at the Institute for Genomic Medicine at the Columbia University Medical Center with 150-bp paired ends. All AstraZeneca clinal trial WES was performed at Human Longevity Inc. with 150-bp paired-ends.

All WES FASTQ data was processed at AstraZeneca using Amazon Web Services cloud computing platform as previously described[23]. Reads were aligned to the GRCh38 genome reference, and small variant calling performed, with the Illumina DRAGEN Bio-IT Platform Germline Pipeline v3.0.7. Variants were annotated with v4.3[47] against Ensembl Build 38.92 and with their genome Aggregation Database (gnomAD) MAFs (gnomAD v2.1.1 mapped to GRCh38)[48].

As previously described[20], the 100,000 Genomes Project was whole genome sequenced using TruSeq DNA polymerase-chain-reaction (PCR)–free sample preparation kit (Illumina) on the HiSeq2500 platform. Reads were aligned using the Isaac Genome Alignment Software, and small variant calling performed with the Platypus variant caller[49]. Variants were annotated with VEP v105 with the gnomAD plugin included[50].

FinnGen genotyping and imputation has been previously described[15]. In brief, genotyping was performed with Illumina (Illumina) and Affymetrix arrays (Thermo Fisher Scientific) and calls with GenCall and zCall algorithms. Imputation was carried out using Beagle 4.1 with a reference panel generated from the WGS of 8554 Finnish individuals (https://finngen.gitbook.io/documentation/methods/genotype-imputation/genotype-imputation). We restricted our analysis to variants within the CCDS region and with imputation INFO ≥ 0.6.

### Cohort harmonisation and quality control

All whole exome sequenced cohorts underwent quality control as previously described[23,51]. Pre-harmonisation and quality control the UKB cohort consisted of 15,417 cases and 147,652 male controls; the MCPS cohort of 287 cases and 46,717 male controls; the NYBAZ cohort of 2506 cases; the clinical trials cohort of 1445 cases. In brief, samples were excluded if contaminated (VerifyBamID contamination ≥ 4%), and if there was discordance between the self-reported and genetically determined sex. Samples were only included for downstream analysis if they achieved ≥94.5% of consensus coding sequence (CCDS) r22 bases covered with ≥10-fold coverage. We excluded participants that were second-degree relatives or closer, estimated with KING v2.2.3[52] using the --kinship function (kinship coefficient > 0.0884). Continent level ancestry was predicted using PEDDY v0.4.2[53] with the 1000 Genomes Project sequences as a population reference. For European cohorts, only individuals with a predicted probability greater than 99% of European ancestry were selected. Non-European strata were included if there were a minimum of 75 cases and the probability threshold was set at greater than 95% for the relevant ancestry. Additionally, only individuals who were within 4 SD of the cohort means for the top four principal components were selected. Finally, samples outside 4 SD of the mean for novel CCDS SNPs in the test cohort were excluded.

For the 100,000 Genomes Project whole genome sequenced cohort, a similar set of harmonisation steps were performed. Before harmonisation the cohort consisted of 1347 cases and 32,985 controls. Pre-harmonisation QC was performed on all whole genome sequences: samples were required not to be contaminated (VerifyBamID free-mix ≤ 3%); aligned reads were required to cover 95% of the genome at 15X or above with mapping quality > 10; array concordance > 90%; median fragment size > 250 bp; chimeric reads < 5%; median fragment size > 250 bp; mapped reads > 60%; AT dropout < 10%; self-reported and genetically determined sex were required to match. For cohort harmonisation, continental ancestry was predicted by training a random forest model on eight 1 kGP3 PCs, and only individuals with a probability of European ancestry greater than 99% were selected. Additionally, only individuals who were within 4 SD of the cohort means for the top four principal components were selected. Finally, participants that were second-degree relatives or closer were removed (prioritising retaining cases), as estimated with KING.

Sample quality control for the FinnGen cohort was as previously described[15], and consisted of ensuring genetically determined sex matched reported sex, low genotype missingness (<5%), and low heterozygosity (±4 standard deviations). Additional cohort harmonisation steps consisted of removing twins/duplicates and those of non-Finnish ancestry (https://finngen.gitbook.io/documentation/methods/phewas/quality-checks).

### Gene-level collapsing analysis

As previously described, we performed gene-level collapsing analysis across eleven QV models[23] (Supplementary Data 1). For dominant collapsing models, carriers with at least one QV were tested against non-carriers. For the single recessive QV model, carriers were defined as those with a homozygous QV, or at least two heterozygous QVs (i.e. putatively compound heterozygous). The association of QV carriers with prostate cancer risk and its severity was tested for with Fisher's exact two-sided test within each cohort. Meta-analysis across cohorts was performed with the Cochran–Mantel–Haenszel (CMH) test. We excluded 56 genes that we previously found to be associated with batch effects. TO DRAGEN WES PASS variant calls we applied additional filters: coverage ≥ 10×; CCDS transcripts annotation (release 22); heterozygous variant alternative allele reads ≥ 0.3 and ≤0.8; alternate allele percentage significantly different from 50% in heterozygous state (binomial $P > 1 \times 10^{-6}$); read position rank sum score (RPRS) ≥ −2; genotype quality score (GQ) ≥ 30; Fisher's strand bias score (FS) ≤ 200 (indels) ≤60

(SNVs); quality score (QUAL) ≥ 30; mapping quality score (MQ) ≥ 40; mapping quality rank sum score (MQRS) ≥ −8; in ≥25% of gnomAD exomes the site achieved 10-fold coverage; if in gnomAD exomes the variant was observed then we required exome z score ≥ −2.0 and exome MQ ≥ 30. For each cohort and each QV model we calculated the genomic inflation factor by regressing observed $P$ values against expected $P$ values generated from n-1 permutation of case-control status (Supplementary Data 2, 6, 10). In the UKB we additionally tested for gene-level association using Firth logistic regression with age, age[2] and four ancestry principal components as covariates to ensure our results were not confounded (Supplementary Data 3 and 7).

### Exome wide association analysis

For next generation sequenced cohorts, single-variant association testing for exome variants was performed as previously described[23]. Variant association with prostate cancer risk and its severity was tested for with Fisher's exact two-sided test under three genetic models: dominant (XX + XY versus YY), allelic (X versus Y) and recessive (XX versus XY + YY), where X is the alternate allele and Y is the ref allele. We applied to DRAGEN WES PASS variant calls additional filters: coverage ≥ 10×; homozygous variant alternative allele reads ≥0.8; heterozygous variant alternative allele reads ≥ 0.3 and ≤0.8; alternate allele percentage significantly different from 50% in heterozygous state (binomial $P > 1 \times 10^{-6}$); FS ≤ 200 (indels) ≤60 (SNVs); MQ ≥ 40; RPRS ≥ −2; QUAL ≥ 30; GQ ≥ 30; MQRS ≥ −8; the variant site does not have less than 10× coverage in 1% or more of sequences; the variant must not have failed any of these QC metrics in >0.5% sequences; in >50% of gnomAD exomes the variant site achieved >10× coverage; Hardy-Weinberg equilibrium test $P < 1 \times 10^{-10}$. Single variant association statistics for the risk of developing prostate cancer in the FinnGen cohort were generated with REGENIE[54] (v2.2.4) under the additive model with sex, age and 10 principal components included as covariates (pipeline details: https://github.com/FINNGEN/regenie-pipelines). Across all cohorts, meta-analysis was performed with the sample sized based (Stouffer's) method as implemented in METAL[55] using allelic or additive summary statistics as available. Additionally for sequenced cohorts, meta-analysis for the dominant and recessive models was performed with CMH.

### Reporting summary

Further information on research design is available in the Nature Portfolio Reporting Summary linked to this article.

## Data availability

For data privacy reasons, individual-level phenotype data and the sequencing data used in this study must be requested directly from each study. Individual-level UK Biobank data may be requested via application to the UK Biobank (www.ukbiobank.ac.uk/register-apply/). Individual-level MCPS data may be requested via Data and Sample Access Policy available on the study's Oxford-hosted webpage (http://www.ctsu.ox.ac.uk/research/mcps). Details on how to access the Genomics England 100,000 Genomes Project individual level data can be found at https://re-docs.genomicsengland.co.uk/pan_cancer_pub/. All FinnGen release r11 association statistics are publicly available (http://r11.finngen.fi). Raw sequencing data used in this study are not available publicly because of privacy protections for the NYBAZ and AZCT cohorts. All other data supporting the findings described in this manuscript are available in the article and its Supplementary Information files. Exome wide summary statistics generated here for genetic association analyses are available on Zenodo (https://doi.org/10.5281/zenodo.14628666).

## Code availability

Gene-level and ExWAS association tests were performed using PEACOK (1.0.7), available on GitHub: https://github.com/astrazeneca-cgr-publications/PEACOK. All other analyses were performed using publicly available software and web-based applications as indicated in the 'Methods' section. Except where specific software packages are named in the 'Methods' section, all statistical analyses and plotting were performed using R (v4.0.4).

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

## Acknowledgements

This research has been conducted using the UK Biobank Resource under Application Number 26041. This research was made possible through access to data in the National Genomic Research Library, which is managed by Genomics England Limited (a wholly owned company of the Department of Health and Social Care). The National Genomic Research Library holds data provided by patients and collected by the NHS as part of their care and data collected as part of their participation in research. The National Genomic Research Library is funded by the National Institute for Health Research and NHS England. The Wellcome Trust, Cancer Research UK and the Medical Research Council have also funded research infrastructure. The generation of the UKB data was funded by the UKB Exome Sequencing Consortium (UKB-ESC) members: AbbVie, Alnylam Pharmaceuticals, AstraZeneca, Biogen, Bristol-Myers Squibb, Pfizer, Regeneron and Takeda. The MCPS has received funding from the Mexican Health Ministry, the National Council of Science and Technology for Mexico, the Wellcome Trust (058299/Z/99), Cancer Research UK, British Heart Foundation and the UK Medical Research Council (MC_UU_00017/2). M.P. received funding from Prostate Cancer Foundation Challenge Award 18CHAL05, NIH/NCI P01 CA228696 and Rebecca and Nathan Milikowsky funded. L.A.M., P.K., K.S., M.P., K.O. and V.J. received funding from the National Cancer Institute (5P01CA228696), and K.O. and V.J. from the Niehaus Center for Inherited Cancer Genomics and the Breast Cancer Research Foundation.

## Author contributions

J.M., K.C., S.P., P.W.K., K.O., L.A.M. and M.P. designed the study. J.M., N.C., P.S., K.H.S., Vijai J., A.O., A.A. and Q.W. performed analyses and statistical interpretation. J.E.B., J.A.D., P.K.M., J.B., R.T.C., J.E., J.M.T, R.C., D.G., A.M., C.H., L.A.D., R.M., Vaidehi J., B.D., S.P., P.W.K., K.O., L.A.M., M.P., M.A.F. contributed to the generation of sequencing and phenotype data. J.M. and M.A.F. wrote the article. N.C., P.S., K.H.S., Vijai J., O.S.B., R.S.D., A.N., J.E.B., A.W.Z., J.E., D.G., A.M., C.H., K.C., S.P., P.W.K., K.O., L.A.M. and M.P. reviewed and edited the article.

## Competing interests

J.M., N.C., O.B., R.D., A.N., A.O., A.A., Q.W., L.A.-D., R.M., B.D., K.C., S.P., M.A.F. are current employees and/or stockholders of AstraZeneca. A.W.Z receives grant funding and consulting fees from AstraZeneca. L.A.M. is on the advisory board and holds equity interest in Convergent Therapeutics. A.M. is a former employee of AstraZeneca and current

employee of GSK and a stockholder of AstraZeneca and GSK. C.H. was an employee and stockholder of AZ at the time of study and is a current employee of Debiopharm International. P.W.K. is a co-founder and employee of Convergent Therapeutics. The remaining authors declare no competing interests.

## Additional information

[1]Centre for Genomics Research, Discovery Sciences, BioPharmaceuticals R&D, AstraZeneca, Cambridge, UK. [2]Institute for Genomic Medicine, Columbia University, New York, NY, USA. [3]Clinical and Translational Epidemiology Unit, Massachusetts General Hospital and Harvard Medical School, Boston, MA, USA. [4]Department of Epidemiology, Harvard T. H. Chan School of Public Health, Boston, MA, USA. [5]Cancer Biology and Genetics Program, Sloan Kettering Institute, New York, NY, USA. [6]Department of Medicine, Memorial Sloan Kettering Cancer Center, New York, NY, USA. [7]Department of Medicine, Weill Cornell Medical College, New York, NY, USA. [8]Department of Molecular and Human Genetics, Baylor College of Medicine, Houston, TX, USA. [9]Dana-Farber Cancer Institute, Boston, MA, USA. [10]Faculty of Medicine, National Autonomous University of Mexico, Copilco Universidad, Coyoacán, Ciudad de México, Mexico. [11]Instituto Tecnológico y de Estudios Superiores de Monterrey, Tecnológico, Monterrey, Nuevo León, Mexico. [12]Clinical Trial Service Unit & Epidemiological Studies Unit, Nuffield Department of Population Health, University of Oxford, Oxford, UK. [13]Centre for Genomics Research, Discovery Sciences, BioPharmaceuticals R&D, AstraZeneca, Waltham, MA, USA. [14]Precision Medicine and Biosamples, R&D Oncology, AstraZeneca, Dublin, Ireland. [15]Department of Pathology and Cell Biology, Columbia University, New York, NY, USA. [16]Oncology R&D, AstraZeneca, Waltham, MA, USA. [17]Convergent Therapeutics, Cambridge, MA, USA. [18]American Cancer Society, Atlanta, GA, USA. [19]Department of Haematology, Cambridge University Hospitals NHS Foundation Trust, Cambridge, UK. [20]Department of Haematology, University of Cambridge, Cambridge, UK. [21]These authors contributed equally: Jonathan Mitchell, Niedzica Camacho. [22]These authors jointly supervised this work: Philip W. Kantoff, Kenneth Offit, Lorelei A. Mucci, Mark Pomerantz. ✉e-mail: jonathan.mitchell@astrazeneca.com; margarete.fabre@astrazeneca.com

