## [Transparent Peer Review file · Nature Communications]

Assessing the contribution of rare protein-coding germline variants to prostate cancer risk and severity in 37,184 cases

Corresponding Author: Dr Jonathan Mitchell

Version 0:

Reviewer comments:

Reviewer #1

(Remarks to the Author)

Mitchell et al report a meta-analysis of prostate cancer risk involving 37,184 prostate cancer cases and 33,1329 controls, from various existing cohorts. They performed a gene-level case-control analysis to assess overall prostate cancer risk, and a case-only analysis to assess risk of aggressive prostate cancer. In a separate single variant-level analysis, they identified variants associated with increased and decreased risk of overall disease. Their analysis of aggressive prostate cancer did not identify any significant association.

The study is well designed with a very large sample size. The analytical design is overall sound, in particular the details that went into the "qualifying variant models" is impressive.

However, this reviewer doesn't think the findings are groundbreaking enough that they warrant publication in this journal. The genes and variants identified to be associated with prostate cancer (overall or aggressive prostate cancer) have already been described. So, although significant due to its size, the study hasn't demonstrated the level of novelty that I would expect to read in this journal.

Minor suggestions: the manuscript would benefit from more thorough proof-reading and English language corrections. Some abbreviations are used before they are defined.

It would be useful to provide the clinically relevant HGVS names for the variants identified.

Reviewer #2

(Remarks to the Author)

Mitchell et al. study associations between prostate cancer and rare germline variants by performing a meta-analysis of some of the world's largest DNA sequencing studies. They did not identify any novel genes associated with prostate cancer, but they found a novel variant in a known gene, and they replicate known associations for several genes.

Overall, this is a large and impressive study. It does not yield any substantial new findings, but it is important for definitively confirming the (less reliable) results of several smaller studies. My main criticism of the manuscript is that the results often seem overstated (see below). For example, the authors do not discuss any limitations of their study.

Major comments:

1. Lines 1-2: The title seems slightly inaccurate to me. I think it is accurate to say the study has characterised the contribution of some rare variants, but certainly not all rare variants or rare variants in general. Some changes should be made to make the title less ambiguous and to avoid these inaccurate interpretations, e.g. "some" could be inserted before "rare variants", or "characterising the contribution of" could be deleted (and some prepositions changed).

2. Lines 46-65: I found the abstract unclear or silent on some critical points. The abstract should mention that the study is a meta-analysis. Also, the abstract should make it clear which results are new, e.g. "revealed" should be changed to "confirmed" or similar.

3. Results (lines 129-299): Currently, the results are phrased in terms of three levels of statistical significance: "study-wide", "suggestive" and "nominal". The first is properly Bonferroni-corrected, and the second is acceptable because the authors appropriately downplay the "suggestive" results. But the nominal level of statistical significance ($p < 0.05$) is essentially meaningless in the context of a large number of hypothesis tests, so it should be removed from the paper. Associations that are currently labelled "nominally significant" should instead be labelled as lacking any evidence for an association.

4. Lines 240-241: The threshold for genomic inflation should be given, and a column should be added to Supplementary Table 11 showing which studies satisfied the genomic inflation criterion. The effect of confounding by ancestry (population stratification) on the study's results should be discussed as a limitation of the study. This is especially important because FinnGen has a genomic inflation factor of 1.13, so it seems that the chosen threshold is not very strict.

5. Line 243: For the variant-level analyses, the number of hypothesis tests performed should be mentioned. It is impossible to interpret the variant-level p-values without this number, e.g. it is impossible to know if the chosen study-wide significance threshold is similar to the Bonferroni threshold for these analyses.

6. Discussion (lines 302-367): A paragraph should be added discussing the study's limitations and their likely impact on the study's results. Ideally, this paragraph would also include perceived limitations. At a minimum, the limitations discussed should include the issue of population stratification (see above) and the use of imputed genotypes in the analyses involving FinnGen data. How much should a reader trust the odds ratio in Table 3 for the novel variant in TERT, given that this is based on imputed data for a very rare variant? Also, does this variant remain significant in a sensitivity analysis where imputed data is excluded? Do any of the variants in Table 3? Even if not, Table 3 can remain, but the study's limitations must be fully acknowledged.

7. Lines 516-520: Some information should be added about how reliable FinnGen's imputation is, especially for rare and very rare variants (such as those in Table 3).

8. Table 3 (lines 881-889): The number of carriers and the number of case-carriers should be added to this table, e.g. in the format n/m in a single column.

Minor comments:

9. Lines 106-111: The example seems to contradict, instead of support, the proposition at the start of the sentence.

10. Results (lines 129-299): Parts of the results seem more like discussion to me, e.g. lines 273-286.

11. Line 157: The acronym "DDR" is not defined until line 180.

12. Lines 161-165: Delete "independently validate the SAMHD1 association with prostate cancer in the UK Biobank". How can this validation be independent, if it is based on the same dataset?

13. Line 230: How can very weak evidence for an association be "strongly suggesting" anything?

14. Lines 298-299: The odds ratio for the TERT variant is 0.134 according to Table 3, so "86%" should be "87%". However, surely there is a better way of describing this finding, e.g. carriers have 7.5-times lower odds than non-carriers.

Reviewer #3

(Remarks to the Author)

Characterising the contribution of rare protein-coding germline variants to prostate cancer risk and severity in 37,184 cases

This is a large, collaborative effort to combine and meta-analyze several datasets with germline sequencing of at least the whole exome. Comparisons are generally between prostate cancer cases and unaffected controls or between aggressive vs. non-aggressive prostate cancer cases. The scope of the study is impressive, especially for sequencing data, and the topic is highly relevant and important.

1. My biggest concern with the study is that the controls are not necessarily well matched to the cases. Several cohorts were specifically recruited as aggressive prostate cancer, making it challenging to find representative controls in a broader population. Using UKB controls for US-based cases is problematic. Controls that were recruited because of disease features for another disease are also not representative of the general population. I suggest repeating each of the main analyses within each cohort and showing the within-cohort results. Obviously, some cohorts use controls from another, but we should still see these results. Though statistical significance may vary with sample size, it would be reassuring to see consistency of point estimates or informative to see the range of point estimates.

2. The abstract should include effect size and significance of the major findings. Significant vs. "suggestive" findings should be distinguished.

3. The novel findings (i.e., not just validation of prior findings) should be highlighted better in the abstract. It appears most of the significant findings are validations of prior discovery (which is fine—but needs to be clear). E.g., the fact that no new DDR genes were identified is notable and useful for the field to know.
4. I am concerned with the lack of covariates in these case-control analyses. For example, why is age not accounted for, if we know that older age at diagnosis is associated with more aggressive disease? Likewise, genetic ancestry and family history are major risk factors that are ignored in this study.
5. The definition of aggressive disease includes administration of chemotherapy and castration resistance. I agree these are aggressive features. What is unknown is how many patients with less aggressive disease might have progressed to castration resistance, higher grade, etc. had they not been treated with radical prostatectomy or radiotherapy. These factors are not addressed in the analyses or the limitations.
6. Lines 323-324: “highlights more broadly the utility of human population genetics for identifying potential therapeutic targets.” I’m not sure I agree. This is an enormous meta-analysis of a very common, highly heritable disease. Finding a potential therapeutic target that we knew about ~1940, 13 years before discovery of the DNA double helix, is not a resounding endorsement of expensive sequencing as a strategy for discovering targets. I suggest reframing this.
7. Lines 356-357: Would soften the language here, as current wording suggests the findings of the study could be predictive of response to treatment rather than just prognosis, but that question has not been investigated here.

Reviewer #4

(Remarks to the Author)

Reviewer #5

(Remarks to the Author)

This paper describes a comprehensive search for rare, exomic risk variants in several large prostate cancer datasets. The authors examine both exome and genome sequencing data, using gene- and variant-level analyses, to validate previously discovered risk variants, in addition to identifying several novel risk variants. While the study is novel and presents new insights into the pathogenesis of prostate cancer, the manuscript requires a little work before being of publication standard.

1. Results section: Table 1 (and most other Tables) require footnotes defining any acronyms present in the table. For example, LCI, UCI and all study acronyms need to be defined.
2. Results section: For Supplementary Figures 1, 4 and 7, I assume the other unlabelled symbols above the expected P-values are results from other QV models for the labelled genes? Could this be clarified/mentioned either in the text or Figure legends, so this is clear.
3. Throughout the Results section are sections of text that are more appropriate to the Discussion section. These are found on p.4 “Germline variants in SAMHD1 have recently been reported...” and the following sentence, p.4 “Although the Duchenne muscular dystrophy (DMD) gene...”, p.5 “Consistent with our association of rare variants in AOX1...”, and the last three paragraphs of the Results section on p.6-7. These sections should be incorporated into the Discussion.
4. While the authors dismiss the DMD and TET2 findings, the results for these genes should still be included in Table 2 similar to the other genes.
5. On p.5, Table 2 should be listed along with Supplementary Tables 8-10 and Supplementary Figures 7-9 (paragraph starting with “Finally, at the gene-level, ...”).
6. On p.5, the last sentence of the same paragraph is confusing as the lower ORs and not the higher ones are mentioned, even though the sentence refers to the larger effect sizes. Either show both for each analysis (e.g., BRCA2 X vs. X) or just refer to Table 2.
7. In the last paragraph on p.5, can the authors specify in the text that the model being referred to is flexdmg. It’s also not clear in Supplementary Table 1 that this model actually includes protein truncating variants, as rare damaging and protein truncating variants are distinct in the models described in the rows above. Can the authors also present the results for all three models being discussed in a Supplementary Table. At present, the reader doesn’t have enough information to determine whether the suggested mechanism of association is valid or not.
8. In the same paragraph as above, can the authors include the DMD and TET2 genes in Supplementary Figure 10? This schematic could be informative, especially for DMD based on the distribution of synonymous variants between cases and controls.
9. Can another column be added to Supplementary Table 11 (p.6) to indicate which studies were/weren’t included in the ExWAS?
10. In the second paragraph of p.6, can the authors provide more information on the eighth locus, e.g., the genes involved, the locus location in the genome or group the 16 variants in Supplementary Table 12 by locus?
11. In the same paragraph on p.6, most of the information contained in the sentence starting with “All seven putatively causal variants...”, is in Figure 3 and/or Supp Table 12. Could the authors add the ORs and CIs to Supplementary Table 12 and just refer to these instead of writing out in the text?
12. Overall, the Discussion needs a lot of work; incorporating sections from the Results section and tightening up some of the assumptions/interpretations made.
13. The first sentence of the Discussion doesn’t really make sense. Do the authors mean the discovery of rare variants deepens our understanding of prostate cancer?
14. The first paragraph of the Discussion ends with “Independently, the genetic variants...” Apart from AOX1, whose association was only suggestive, I’m not sure the authors can say the discovered rare variants independently play a role in

cancer severity. BRCA2 maybe, but an association of BRCA2 in the non-aggressive cases vs. controls would need to be determined in order to be more certain. Based on the study's findings, I find it hard to accept that ATM plays an independent role.

15. Second paragraph of the Discussion, "Beyond the DDR, we identify...", should perhaps be changed to "we validate or identify rare non-synonymous variants..." given variants in ANO7 and HOXB13 have previously been identified.

16. I find the last point of the above paragraph a little confusion. Do the authors mean the utility of pop. genetics is to identify men who may benefit from certain therapies or is the utility to inform therapeutic strategies? If the variant in AR is protective, it could be argued that you wouldn't want to give men ADT, as this would diminish AR levels and potentially protection.

However, this isn't a gene that is associated with severe prostate cancer, so it's therapeutic potential isn't great regardless. I feel that these variants are more suited to developing better screening strategies.

17. The paragraph discussing SAMDH1 (p.6-7) is also a little confusing/disjointed. It needs to be made clear that the same QV model found an association between SAMHD1 and telomere length, as it currently reads as though just the model was the same. Shorter telomers are also more often associated with increased cancer risk, not reduced. While I realise the opposite has also been suggested, I'd suggest reading up a little more on the association between telomers and cancer risk (e.g., Okamoto and Seimiya have a review paper) and revising this paragraph.

18. The last sentence of the Discussion mentions prostate cancer prevention, how has this study provided evidence for therapeutic approaches to prevention? Do the authors mean their work has implications for distinct screening and therapeutic approaches?

19. In the Methods section (second paragraph), is it correct that the MCPS cohort has had no phenotype updates, whether through linkage to a cancer registry or death records since 2004? Or was the linkage to death registry data done more recently? Can this be clarified?

Version 1:

Reviewer comments:

Reviewer #2

(Remarks to the Author)

The authors have addressed all of my main concerns, and their changes have not introduced any new issues. The only remaining issues are extremely minor and, I think, can be left to the discretion of the authors.

In the numbering of my initial review:

4. I still think the failure to account for population stratification could be acknowledged more fully as a limitation of the paper. This issue seems to me just as applicable for sequencing studies as GWASs, but more problematic for sequencing studies because it's harder to adjust for ancestry. If a variant is only present in people from Northern Finland, and Northerners are more likely to develop prostate cancer than Southerners due to unrelated lifestyle differences, then wouldn't you say this rare variant is associated with prostate cancer? Though I accept that this issue is similar to the more important issue of cases and controls being drawn from different populations, which you acknowledge.

8. For the TERT variant, all counts were NA. Should these be zero instead?

Reviewer #3

(Remarks to the Author)

Most of the responses to my comments are reassuring, especially the forest plots.

I agree that the unknowns regarding aggressive disease would bias toward the null. Given that this study did not discover many (any?) novel variants, the concern remains. The authors have now acknowledged this limitation, which is adequate.

My remaining concern is regarding covariates. If I understood it correctly, the authors' response explains that Fisher's exact test is computationally efficient and has been useful in other publications/studies. That is all fine, but I am not sure it answers my question. If there is an imbalance between cases and controls in a known confounder (e.g., age, genetic ancestry, or family history), this could yield misleading results for genetic associations.

Reviewer #4

(Remarks to the Author)

Reviewer #5

(Remarks to the Author)

The Authors have made substantial changes to the manuscript and their description of the results based on the Reviewers'

comments. As such, the manuscript is much improved and more clearly reflects the data presented.

Response to Reviewers

Assessing the contribution of rare protein-coding germline variants to prostate cancer risk and severity in 37,184 cases

We are grateful to the editorial team and reviewers for their constructive comments and for the opportunity to submit a revised manuscript. We would like to highlight some key areas of improvement:

1. We have edited the manuscript to improve the clarity and completeness of our interpretation of results. This includes, for example, the addition of a paragraph in the Discussion section outlining the study's limitations, and improved distinction between findings that are novel and those that replicate previous discoveries.
2. We have addressed the potential issue of including studies where the cases and controls were recruited from separate cohorts, and now discuss the steps taken to mitigate false positives (bioinformatics pipeline homogeneity and careful cohort harmonisation). Reassuringly, for all gene-level associations observed in the meta-analysis, the associations were also evident in the within-cohort UK Biobank (UKB) analysis. This is now more explicit in the revised manuscript, including the addition of new Supplementary Figures 2,7 & 11.

Collectively, we feel that the revisions have improved the overall quality of our work and its reporting and thank the reviewers for their feedback.

Below, we provide point-by-point responses, with reviewer comments in blue font and our responses in black font. All edits in the manuscript are highlighted in red.

REVIEWER COMMENTS

Reviewer #1 (Remarks to the Author):

Mitchell et al report a meta-analysis of prostate cancer risk involving 37,184 prostate cancer cases and 33,1329 controls, from various existing cohorts. They performed a gene-level case-control analysis to assess overall prostate cancer risk, and a case-only analysis to assess risk of aggressive prostate cancer. In a separate single variant-level analysis, they identified variants associated with increased and decreased risk of overall disease. Their analysis of aggressive prostate cancer did not identify any significant association.

The study is well designed with a very large sample size. The analytical design is overall sound, in particular the details that went into the "qualifying variant models" is impressive.

However, this reviewer doesn't think the findings are groundbreaking enough that they warrant publication in this journal. The genes and variants identified to be associated with prostate cancer (overall or aggressive prostate cancer) have already been described. So, although significant due to its size, the study hasn't demonstrated the level of novelty that I would expect to read in this journal.

Minor suggestions: the manuscript would benefit from more thorough proof-reading and English language corrections. Some abbreviations are used before they are defined. It would be useful to provide the clinically relevant HGVS names for the variants identified.

We thank the Reviewer for reviewing our manuscript and for highlighting that the study is analytically well designed and powered by a very large sample size. While we recognise their concern regarding the novelty of the findings, we believe our study makes several important contributions to the field. By bringing together sequencing data from biobanks, prostate cancer specific cohorts and clinical trial participants, we have performed the largest study to date analysing the association of rare variants with prostate cancer. This has enabled the confirmation of prostate cancer gene associations identified in smaller single cohort studies (for example the recently reported *SAMHD1*, PMID: 39261734), and the identification of novel associations at the study wide level (*TERT*) and at the suggestive level (*AOX1*) of significance. Our robust confirmation of reported genes, and discovery of novel associations, have the potential to provide valuable insights into the pathogenesis of prostate cancer. In addition, as the largest study of its kind, our results provide a useful reference for genes which were *not* found to be significantly associated with prostate cancer (for example, certain components of the DNA damage response pathway).

Overall, we believe that this study will provide valuable insights and represent an important resource to the community. Its contents are similar in nature to previous

whole exome sequencing studies of cancer, with examples published in *Nature Genetics* (PMID: 37592023) and *Nature Communications* (PMID: 27329137).

We also thank the Reviewer for the minor suggestions on improving the manuscript. We have performed careful proof-reading to correct English language errors, ensured timely use of abbreviations, and added the HGVS variant names to Table 3.

Reviewer #2 (Remarks to the Author):

Mitchell et al. study associations between prostate cancer and rare germline variants by performing a meta-analysis of some of the world's largest DNA sequencing studies. They did not identify any novel genes associated with prostate cancer, but they found a novel variant in a known gene, and they replicate known associations for several genes.

Overall, this is a large and impressive study. It does not yield any substantial new findings, but it is important for definitively confirming the (less reliable) results of several smaller studies. My main criticism of the manuscript is that the results often seem overstated (see below). For example, the authors do not discuss any limitations of their study.

We thank the reviewer for their positive comments and the recognition of the importance of our study. The comments on how to improve the reporting of our study are appreciated. As detailed below, we have addressed all comments, including the addition of a section in the Discussion on the limitations of the study.

Major comments:

1. Lines 1-2: The title seems slightly inaccurate to me. I think it is accurate to say the study has characterised the contribution of some rare variants, but certainly not all rare variants or rare variants in general. Some changes should be made to make the title less ambiguous and to avoid these inaccurate interpretations, e.g. "some" could be inserted before "rare variants", or "characterising the contribution of" could be deleted (and some prepositions changed).

To better reflect the analysis performed and the findings, we have replaced "characterising" with "assessing" in the title.

2. Lines 46-65: I found the abstract unclear or silent on some critical points. The abstract should mention that the study is a meta-analysis. Also, the abstract should make it clear which results are new, e.g. "revealed" should be changed to "confirmed" or similar.

To make it explicit in the abstract that we are performing a meta-analysis, we have added the phrase "performed a meta-analysis on". In addition, we have clarified in the abstract which results are new to this study with "revealed", and those which we replicate in this larger study with "confirmed".

3. Results (lines 129-299): Currently, the results are phrased in terms of three levels of statistical significance: "study-wide", "suggestive" and "nominal". The first is properly Bonferroni-corrected, and the second is acceptable because the authors appropriately downplay the "suggestive" results. But the nominal level of statistical significance ($p < 0.05$) is essentially meaningless in the context of a large number of hypothesis tests, so it should be removed from the paper. Associations that are currently labelled "nominally

significant” should instead be labelled as lacking any evidence for an association.

We agree with the reviewer that the term “nominal significance” was not well defined, and believe the changes we have made to address this comment improve the clarity of our interpretation. In all cases where the term “nominal” significance was used, it has been removed and we now alternatively report the statistics directly:

“We found no significant association with any additional DDR genes (Supplementary Figure 4 and Supplementary Table 5). However, this could be due to a lack of power related to low carrier frequency; in *MSH2*, for example, it is notable that the effect size estimate for PTVs (OR=3.38 [1.55-6.90], $P=1.20 \times 10^{-3}$) was similar to the effect sizes in DDR genes found to be significantly associated with the overall risk of prostate cancer.”

“Beyond *BRCA2*, of the genes found to be associated with prostate cancer risk, the DDR gene *ATM* showed the strongest evidence of also being associated with severity (OR=2.23 [1.47-3.34], $P=9.41 \times 10^{-5}$, Figure 2, Supplementary Figure 8 and Supplementary Table 9).”

“To assess whether these associations were also operating via a loss-of-function mechanism, we looked for evidence in the QV model that includes only PTVs (“ptv”), and observed associations at $P < 0.05$ in a consistent direction in all three cases (*CHEK2*, OR=1.58 [1.00-2.41], $P=0.035$; *SAMHD1*, OR=2.15 [1.22-3.63], $P=0.006$; *AOX1*, OR=3.63 [1.36-9.63], $P=0.006$, Supplementary Tables 4 & 8).”

“Although the significantly associated *BIK* conservative inframe deletion variant was unique to the FinnGen cohort, a separate rare disruptive inframe deletion in the same gene was present in UKB (22:43129228:GTGCTGCTGGCGCTGCTGC:G, OR=1.49 [1.20-1.85], $P=6.20 \times 10^{-4}$).”

4. Lines 240-241: The threshold for genomic inflation should be given, and a column should be added to Supplementary Table 11 showing which studies satisfied the genomic inflation criterion. The effect of confounding by ancestry (population stratification) on the study’s results should be discussed as a limitation of the study. This is especially important because FinnGen has a genomic inflation factor of 1.13, so it seems that the chosen threshold is not very strict.

We thank the reviewer for pointing out the omission of the genomic inflation factor threshold ($\lambda < 1.15$). This is now stated in the Results section (line 234) and a column has been added to Supplementary Table 14 (previously Supplementary Table 11) to indicate whether the study was included in the ExWAS meta-analysis.

Prostate cancer is highly heritable and polygenic, and therefore we believe that an inflation factor of 1.13 is not excessive in the context of polygenicity (PMID: 25642630) and the large sample size of the FinnGen study (17,258 cases and 143,624 controls). Additionally, QC steps were taken to remove outliers based on ancestry (<https://finngen.gitbook.io/documentation/methods/phewas/quality-checks>) and genetic association testing was performed with REGENIE which accounts for

population structure (PMID 34017140). However, we appreciate the reviewer's concern regarding population stratification induced inflation and have now raised this in the Discussion in the context of studies where the cases and controls were recruited from separate cohorts:

“Firstly, the gene-level association meta-analysis includes studies where the cases and the controls were recruited from separate cohorts. While this is a necessary approach for including disease specific cohorts in rare variant association studies, biases may be introduced from technical artefacts and population differences (PMID: 35581355, 33326012). In this study, we mitigated these potential biases by using the same bioinformatics pipeline for cases and controls, and by using strict quality control criteria aimed at ensuring cohort harmonisation. Although it is not possible to entirely rule out that some bias remained, we did not observe significant genomic inflation in our association test statistics, and reassuringly there was evidence from multiple cohorts for all statistically significant associations.”

5. Line 243: For the variant-level analyses, the number of hypothesis tests performed should be mentioned. It is impossible to interpret the variant-level p-values without this number, e.g. it is impossible to know if the chosen study-wide significance threshold is similar to the Bonferroni threshold for these analyses.

We have clarified this in the results section:

“We tested 1,573,300 variants using three genetic models (additive, dominant and recessive, see Methods), and set a threshold of $P < 1 \times 10^{-8}$ for study-wide statistical significance (PMID 34375979)”.

The threshold we have used is therefore slightly more strict than a Bonferroni threshold, corrected over all three genetic models ($P = 0.05 / (3 \times 1573300) = 1.06 \times 10^{-8}$).

6. Discussion (lines 302-367): A paragraph should be added discussing the study's limitations and their likely impact on the study's results. Ideally, this paragraph would also include perceived limitations. At a minimum, the limitations discussed should include the issue of population stratification (see above) and the use of imputed genotypes in the analyses involving FinnGen data. How much should a reader trust the odds ratio in Table 3 for the novel variant in TERT, given that this is based on imputed data for a very rare variant? Also, does this variant remain significant in a sensitivity analysis where imputed data is excluded? Do any of the variants in Table 3? Even if not, Table 3 can remain, but the study's limitations must be fully acknowledged.

We agree that population stratification and imputed genotypes are potential limitations of the study, and have now raised them in the discussion section. Specifically for the FinnGen study, we do not believe population stratification to be problematic for the reasons stated in response to comment 4. However, we have addressed possible confounding in the sequenced cohorts in the Discussion, as quoted in our response to comment 4.

Regarding the imputed genotypes in FinnGen, although the significantly associated single variants in Table 3 are rare in the general European population, they are enriched in the Finnish population. We have now included minor allele frequency (MAF) in Finnish Europeans in Table 3 to show this. This makes the imputation of these variants from the Finnish reference panel more reliable and all significantly associated variants reported have INFO>0.7 (imputation INFO score now added to Supplementary Table 15). However, we still recognise the limitation of the imputed data and have included this point in the Discussion:

“Thirdly, FinnGen’s genotyping data is imputed and, although the imputation utilised a population-specific reference panel of high-coverage WGS data (PMID: 36653562) and we excluded low quality imputed variants, findings derived from imputed variants should be interpreted with greater caution than those derived from direct sequencing.”

Additionally, we have now summarised the evidence for association beyond the FinnGen cohort in the Results section:

“In FinnGen, all seven variants were significantly associated with prostate cancer risk ($P<1\times 10^{-8}$), and for four of the variants there was evidence of association ($P<0.05$) after excluding FinnGen and meta-analysing the sequenced cohorts alone (Supplementary Table 15).”

Collectively, we believe these changes improve the interpretation of the results, while also acknowledging the limitations, and we thank the reviewer for raising the issue.

7. Lines 516-520: Some information should be added about how reliable FinnGen’s imputation is, especially for rare and very rare variants (such as those in Table 3).

We thank the reviewer for highlighting the omission of additional information on FinnGen imputation, the inclusion of which we feel provides greater confidence in our results. In the Methods section (lines 521-522) we now state that we only included imputed variants with INFO \geq 0.6, and also added the INFO score to the significantly associated variants listed in Supplementary Table 15, all of which had INFO>0.7.

In addition, as discussed in our reply to Comment 6, we note that while variants in Table 3 are rare in non-Finnish Europeans, they are enriched in the Finnish population. To show this, we have now added a column to Table 3 with the Finnish population frequency for the variants.

8. Table 3 (lines 881-889): The number of carriers and the number of case-carriers should be added to this table, e.g. in the format n/m in a single column.

To provide the reader with carrier number information, we have added the number of alternative alleles in cases and controls to Supplementary Table 15 for each of the sequenced cohorts, and also the case and control alternative allele frequencies in the array imputed FinnGen cohort.

Minor comments:

9. Lines 106-111: The example seems to contradict, instead of support, the proposition at the start of the sentence.

This sentence has now been edited for clarity, and an additional reference cited: “Importantly, emerging evidence suggests that the set of genes influencing the risk of developing prostate cancer is, at least in part, distinct from genes influencing prostate cancer aggressiveness (<https://doi.org/10.1101/2023.10.10.23296544>, PMID: 37945903). For example, a genetic risk score incorporating disease risk variants was not associated with severity in men of European, Asian and Hispanic ancestries, and did so only modestly in men of African ancestry, suggesting that additional genetic variants, not captured by the genetic score for risk of disease development, might influence disease behaviour (PMID: 37945903)”

10. Results (lines 129-299): Parts of the results seem more like discussion to me, e.g. lines 273-286.

We thank the reviewer for the suggestion on improving the structure and flow of the manuscript. We have re-worked the Discussion, which has involved moving sections of Results to Discussion, including what were lines 273-286 (now lines 312-329). All changes to the Results/Discussion are highlighted in the manuscript in red.

11. Line 157: The acronym “DDR” is not defined until line 180.

We apologise for this oversight. We have now moved the definition of DDR to its first use in the manuscript (line 158).

12. Lines 161-165: Delete “independently validate the SAMHD1 association with prostate cancer in the UK Biobank”. How can this validation be independent, if it is based on the same dataset?

We have edited the sentence in the manuscript to now state that we “reproduce” as opposed to “validate” the association of SAMHD1 with prostate cancer in the UKB (line 280).

13. Line 230: How can very weak evidence for an association be “strongly suggesting” anything?

We have expanded and edited this section so that the interpretation of results is in line with the evidence. Specifically:

- We have now highlighted the observation that the qualifying variant (QV) model most significantly associated with prostate cancer risk and severity in *BRCA2* and *ATM* was the ‘ptv model’ (containing only protein-truncating variants (PTVs)),

suggesting that these genes operate via a loss-of-function mechanism in prostate cancer.

- We have included two observations relating to QV model type for *CHEK2* / *SAMHD1* (for disease risk) and *AOX1* (for disease severity).
 - (i) The most significant QV model for each of these genes was the “flexdmg” model, which includes a combination of rare predicted damaging missense and protein-truncating variants, as specified in Table 2 [*CHEK2*, OR=1.69 [1.41-2.01], $P=2.69 \times 10^{-8}$; *SAMHD1*, OR=2.02 [1.65-2.45], $P=2.36 \times 10^{-11}$; *AOX1* OR=2.60 [1.75-3.83], $P=1.35 \times 10^{-6}$].
 - (ii) For each of these three genes, the PTV-only model showed associations at $P < 0.05$ in a consistent direction in all three cases (*CHEK2*, OR=1.58 [1.00-2.41], $P=0.035$; *SAMHD1*, OR=2.15 [1.22-3.63], $P=0.006$; *AOX1*, OR=3.63 [1.36-9.63], $P=0.006$).

Taken together, we believe that this constitutes sufficient evidence to suggest that these genes might exert their effect in prostate cancer via loss-of-function.

We have included the following text to reflect the above:

“Leveraging variant type to infer direction of effect, our observation that the QV model most significantly associated with prostate cancer risk and severity in *BRCA2* and *ATM* was the ‘ptv model’ (containing only PTVs), suggests that these genes operate via a loss-of-function mechanism in prostate cancer (Table 2, Supplementary Table 1). For three additional genes – *CHEK2*, *SAMHD1* and *AOX1* – the most significant QV model included a combination of rare predicted damaging missense and protein-truncating variants (“flexdmg”, Table 2, Supplementary Table 1). To assess whether these associations were also operating via a loss-of-function mechanism, we looked for evidence in the QV model that includes only PTVs (“ptv”), and observed associations at $P < 0.05$ in a consistent direction in all three cases (*CHEK2*, OR=1.58 [1.00-2.41], $P=0.035$; *SAMHD1*, OR=2.15 [1.22-3.63], $P=0.006$; *AOX1*, OR=3.63 [1.36-9.63], $P=0.006$, Supplementary Tables 4 & 8).”

14. Lines 298-299: The odds ratio for the TERT variant is 0.134 according to Table 3, so “86%” should be “87%”. However, surely there is a better way of describing this finding, e.g. carriers have 7.5-times lower odds than non-carriers.

We thank the reviewer for the suggestion on how to improve this description and have implemented it in the revised manuscript: “carriers have a 7.5-times lower odds compared to non-carriers of developing prostate cancer”.

Reviewer #3 (Remarks to the Author):

Characterising the contribution of rare protein-coding germline variants to prostate cancer risk and severity in 37,184 cases

This is a large, collaborative effort to combine and meta-analyze several datasets with germline sequencing of at least the whole exome. Comparisons are generally between prostate cancer cases and unaffected controls or between aggressive vs. non-aggressive prostate cancer cases. The scope of the study is impressive, especially for sequencing data, and the topic is highly relevant and important.

We thank the Reviewer for the positive review of the manuscript, highlighting the importance of the study, and for providing areas for improvement. All comments have been addressed below and edits made accordingly, which we believe has strengthened our manuscript.

1. My biggest concern with the study is that the controls are not necessarily well matched to the cases. Several cohorts were specifically recruited as aggressive prostate cancer, making it challenging to find representative controls in a broader population. Using UKB controls for US-based cases is problematic. Controls that were recruited because of disease features for another disease are also not representative of the general population. I suggest repeating each of the main analyses within each cohort and showing the within-cohort results. Obviously, some cohorts use controls from another, but we should still see these results. Though statistical significance may vary with sample size, it would be reassuring to see consistency of point estimates or informative to see the range of point estimates.

We agree with the reviewer that this could be a potential limitation of the study and have now raised it in the Discussion section along with the steps we have taken to mitigate false positives, and evidence to support the robustness of our results:

“Firstly, the gene-level association meta-analysis includes studies where the cases and the controls were recruited from separate cohorts. While this is a necessary approach for including disease specific cohorts in rare variant association studies, biases may be introduced from technical artefacts and population differences (PMID: 35581355, 33326012). In this study, we mitigated these potential biases by using the same bioinformatics pipeline for cases and controls, and by using strict quality control criteria aimed at ensuring cohort harmonisation. Although it is not possible to entirely rule out that some bias remained, we did not observe significant genomic inflation in our association test statistics, and reassuringly there was evidence from multiple cohorts for all statistically significant associations.”

For all associations found to be significant in the meta-analysis at the gene-level, the results for each individual cohort are detailed (Supplementary Tables 3, 7 and 11), regardless of whether the controls were derived from a separate cohort (as was the case for New York-Boston-AstraZeneca and AstraZeneca clinical trial cohorts) or from

within the same cohort (all others, with details of control selection in Methods lines 462-477). We have also generated new Supplementary Figures (2, 7 & 11), consisting of forest plots showing these point estimates for each cohort. Reassuringly, we observed that for all gene-level associations reported (for both risk and for disease severity) there was evidence for the association in the within-cohort UKB analysis at $P < 0.05$ (Supplementary Tables 3 and 7).

2. The abstract should include effect size and significance of the major findings. Significant vs. “suggestive” findings should be distinguished.

We thank the Reviewer for these suggestions and agree that they will make the abstract clearer. To this end:

- We now qualify ‘significant’ and ‘suggestive’ findings, quoting the P-value thresholds for each ($P < 1 \times 10^{-8}$ & $P < 2.6 \times 10^{-6}$ respectively).
- We include the absolute effect sizes and P-values of the novel associations.

3. The novel findings (i.e., not just validation of prior findings) should be highlighted better in the abstract. It appears most of the significant findings are validations of prior discovery (which is fine—but needs to be clear). E.g., the fact that no new DDR genes were identified is notable and useful for the field to know.

We now clarify this in the abstract by using the terms “revealed” and “novel” for new findings, and “confirmed” for validation of previously-described findings. We agree that there is substantial importance in the absence of additional novel genes, particularly in the DDR pathway, and we now highlight this in the abstract.

4. I am concerned with the lack of covariates in these case-control analyses. For example, why is age not accounted for, if we know that older age at diagnosis is associated with more aggressive disease? Likewise, genetic ancestry and family history are major risk factors that are ignored in this study.

We are grateful for the opportunity to address the question of covariates, which was carefully considered when designing our study. Firstly, to avoid possible confounding in the genetic association analyses we performed strict cohort harmonisation (Methods: lines 524-563), which included multiple steps to ensure cases and controls were well matched for genetic ancestry, and as a result we do not observe significant genomic inflation. Secondly, regarding factors which are associated with outcome but are not confounders (e.g. age), we have previously shown that our approach using Fisher’s exact test performs well when compared to a generalized mixed model approach (as implemented in SAIGE, PMID 30104761) including covariates (PMID 34375979). In particular, our previous work comparing analytic approaches has demonstrated that, with our cohort harmonisation steps: (1) we observe a near-perfect correlation between phred scores for significant associations from the exact test and SAIGE, (2) Fisher’s exact test allows robust assessment of association for very rare variants and (3) our exact based method is significantly more computationally efficient (as we described previously, PMID 34375979 response to Referee 1 Comment 3 here). As a result, at AstraZeneca’s Centre for Genomics Research, we have successfully used the exact test

based approach in a number of large-scale sequencing-based association studies, including those published in *Nature*, *Science* and *JAMA Cardiology* (PMIDs 34375979, 37794183, 25700176 and 33326012). We believe that the above constitutes a sound basis for implementing a more sparse-observation robust exact test implementation for the current study that focuses primarily on exploring the contributions of rare genetic variants.

5. The definition of aggressive disease includes administration of chemotherapy and castration resistance. I agree these are aggressive features. What is unknown is how many patients with less aggressive disease might have progressed to castration resistance, higher grade, etc. had they not been treated with radical prostatectomy or radiotherapy. These factors are not addressed in the analyses or the limitations.

We thank the reviewer for raising this point. There is no doubt that therapeutic interventions will alter the natural history of disease, in particular preventing progression and thereby potentially causing under-estimation of aggressiveness in some cases. Indeed, it is likely that the natural biology in a proportion of cases classified as ‘non-aggressive’ in our study is, in fact, aggressive. Importantly however, we do not believe that this mis-classification would have introduced false positive genetic associations, but would instead have reduced our power to detect true signals. An analogous potential dilution of signal is potentially also operating in our case-control analysis, where individuals unknowingly destined to go on to develop prostate cancer in future are used as controls.

We have included additional text in the Discussion to reflect this:

“the theoretical misclassification of (i) controls (which might have included individuals unknowingly destined to develop prostate cancer in future) and (ii) non-aggressive cases (which might have included individuals who would have developed features of aggressive disease had they not received treatment), potentially reduced our power to detect genetic signals.”

6. Lines 323-324: “highlights more broadly the utility of human population genetics for identifying potential therapeutic targets.” I’m not sure I agree. This is an enormous meta-analysis of a very common, highly heritable disease. Finding a potential therapeutic target that we knew about ~1940, 13 years before discovery of the DNA double helix, is not a resounding endorsement of expensive sequencing as a strategy for discovering targets. I suggest reframing this.

We are grateful for the reviewer’s comment on this; we had not intended to use the *AR* finding to imply the merit of large-scale sequencing over other methods for drug target discovery. However, we believe that the association of prostate cancer with germline variants in the androgen receptor pathway, a well established therapeutic target in prostate cancer, serves as proof of the principle that human population genetics can be leveraged for target discovery. As such, in the current analysis, *AR* could be regarded as a positive control, suggesting that the other genes found to be associated with prostate

cancer risk/severity could be considered, alongside other strands of evidence, as potential therapeutic targets.

We have reframed the relevant sentence in the manuscript to reflect this:

“Finally, the protective missense variant in *AR*, which encodes the androgen receptor, is notable given the widespread treatment of prostate cancer patients with anti-androgen therapies (PMID: 19796750), and highlights the connection between rare germline variant disease associations and potential therapeutic targets.”

7. Lines 356-357: Would soften the language here, as current wording suggests the findings of the study could be predictive of response to treatment rather than just prognosis, but that question has not been investigated here.

The language has been edited to reflect that the clinical implications of our findings are conjecture that require further study:

“Similarly, inheritance of variants associated with aggressive prostate cancer could impact intensity of monitoring and/or treatment decisions. Both of these projections require further investigation in dedicated studies.”

Reviewer #4 (Remarks to the Author):

I co-reviewed this manuscript with one of the reviewers who provided the listed reports. This is part of the NatureCommunications initiative to facilitate training in peer review and to provide appropriate recognition for Early Career Researchers who co-review manuscripts.

We appreciate the Reviewer taking the time to co-review our manuscript.

Reviewer #5 (Remarks to the Author):

This paper describes a comprehensive search for rare, exomic risk variants in several large prostate cancer datasets. The authors examine both exome and genome sequencing data, using gene- and variant-level analyses, to validate previously discovered risk variants, in addition to identifying several novel risk variants. While the study is novel and presents new insights into the pathogenesis of prostate cancer, the manuscript requires a little work before being of publication standard.

We thank the Reviewer for the thorough review of our manuscript and for highlighting the novelty of the study and the insights generated. All comments have been addressed in the new version of the manuscript, as described in detail below. We believe that these revisions address the Reviewer's concerns and we thank them for their feedback.

1. Results section: Table 1 (and most other Tables) require footnotes defining any acronyms present in the table. For example, LCI, UCI and all study acronyms need to be defined.

For all Tables and Supplementary Tables we have now defined all acronyms in the table descriptions.

2. Results section: For Supplementary Figures 1, 4 and 7, I assume the other unlabelled symbols above the expected P-values are results from other QV models for the labelled genes? Could this be clarified/mentioned either in the text or Figure legends, so this is clear.

Yes, this is correct and in each of the Figure legends we have written "Genes which reach the suggestive significance threshold ($P < 2.6 \times 10^{-6}$) are labelled, and only the most significant qualifying variant model for each gene is labelled."

3. Throughout the Results section are sections of text that are more appropriate to the Discussion section. These are found on p.4 "Germline variants in SAMHD1 have recently been reported..." and the following sentence, p.4 "Although the Duchenne muscular dystrophy (DMD) gene...", p.5 "Consistent with our association of rare variants in AOX1...", and the last three paragraphs of the Results section on p.6-7. These sections should be incorporated into the Discussion.

The Discussion section has been extensively rewritten, and now incorporates sections which were previously in Results. The sections moved from Results to Discussion include the three examples given in the Reviewer's comment (now lines 280, 290 and 324-327).

4. While the authors dismiss the DMD and TET2 findings, the results for these genes should still be included in Table 2 similar to the other genes.

These genes and their association statistics have now been added to Table 2.

5. On p.5, Table 2 should be listed along with Supplementary Tables 8-10 and Supplementary Figures 7-9 (paragraph starting with “Finally, at the gene-level, ...”).

Table 2 is now listed along with the Supplementary Tables and Figures at the end of the sentence:

“Finally, at the gene-level, we tested for genetic association between aggressive prostate cancer and controls, and found that PTVs in *BRCA2* (OR=8.23 [6.17-10.85], $P=1.47 \times 10^{-36}$) and *ATM* (OR=5.27 [3.65-7.46], $P=1.74 \times 10^{-16}$) were significantly associated with aggressive disease (Table 2, Supplementary Tables 10-13, Supplementary Figures 9-12).”

6. On p.5, the last sentence of the same paragraph is confusing as the lower ORs and not the higher ones are mentioned, even though the sentence refers to the larger effect sizes. Either show both for each analysis (e.g., *BRCA2* X vs. X) or just refer to Table 2.

For clarity, we now simply refer to Table 2 at the end of the sentence, as suggested:

“Consistent with *BRCA2* and *ATM* showing association with disease severity, their effect sizes were larger in this aggressive prostate cancer versus controls analysis compared to the overall prostate cancer versus controls analysis (Figure 2 and Table 2).”

7. In the last paragraph on p.5, can the authors specify in the text that the model being referred to is flexdmg. It’s also not clear in Supplementary Table 1 that this model actually includes protein truncating variants, as rare damaging and protein truncating variants are distinct in the models described in the rows above. Can the authors also present the results for all three models being discussed in a Supplementary Table. At present, the reader doesn’t have enough information to determine whether the suggested mechanism of association is valid or not.

We thank the reviewer for these suggestions. We addressed the points by doing the following:

- We have now stated in the manuscript text that it is the “flexdmg” model being referred to (line 219).
- We have modified the footnote describing the variant types for the QV models in Supplementary Table 1, making it clearer that “flexdmg” includes both missense and protein truncating variants.
- We have included three additional Supplementary Tables (4, 8 and 12), giving full details of the association statistics for all QV models in the case vs control, aggressive vs non-aggressive, and aggressive vs controls analyses.
- We have edited the manuscript text to clarify the evidence behind the suggested mechanisms of association, as follows:

“Leveraging variant type to infer direction of effect, our observation that the QV model most significantly associated with prostate cancer risk and severity in *BRCA2* and *ATM* was the ‘ptv model’ (containing only PTVs), suggests that these genes operate via a loss-of-function mechanism in prostate cancer (Table 2, Supplementary Table 1). For three

additional genes – *CHEK2*, *SAMHD1* and *AOX1* – the most significant QV model included a combination of rare predicted damaging missense and protein-truncating variants (“flexdmg”, Table 2, Supplementary Table 1). To assess whether these associations were also operating via a loss-of-function mechanism, we looked for evidence in the QV model that includes only PTVs (“ptv”), and observed associations at $P < 0.05$ in a consistent direction in all three cases (*CHEK2*, OR=1.58 [1.00-2.41], $P=0.035$; *SAMHD1*, OR=2.15 [1.22-3.63], $P=0.006$; *AOX1*, OR=3.63 [1.36-9.63], $P=0.006$, Supplementary Tables 4 & 8.)”

8. In the same paragraph as above, can the authors include the *DMD* and *TET2* genes in Supplementary Figure 10? This schematic could be informative, especially for *DMD* based on the distribution of synonymous variants between cases and controls.

Lollipop plots have now been generated for *TET2* and *DMD* and added to Supplementary Figure 13 (previously Supplementary Figure 10).

9. Can another column be added to Supplementary Table 11 (p.6) to indicate which studies were/weren’t included in the ExWAS?

We have added a column to Supplementary Table 14 (previously Supplementary Table 11) to indicate which studies were included in the ExWAS meta-analysis.

10. In the second paragraph of p.6, can the authors provide more information on the eighth locus, e.g., the genes involved, the locus location in the genome or group the 16 variants in Supplementary Table 12 by locus?

We have added a column to Supplementary Table 15 (previously Supplementary Table 12) indicating the locus to make it easier for the reader to identify which variants are in the same locus. In combination with the posterior inclusion probability, this makes clear which variant is the lead variant in each locus.

11. In the same paragraph on p.6, most of the information contained in the sentence starting with “All seven putatively causal variants...”, is in Figure 3 and/or Supp Table 12. Could the authors add the ORs and CIs to Supplementary Table 12 and just refer to these instead of writing out in the text?

We have edited the text to remove the statistics and instead refer the reader to Table 2 and Supplementary Table 15 (previously Supplementary Table 12), as suggested.

12. Overall, the Discussion needs a lot of work; incorporating sections from the Results section and tightening up some of the assumptions/interpretations made.

The Discussion section has been substantially reworked and now incorporates large parts of what was previously in the Results section (all manuscript edits are highlighted in red). The rewriting also involved edits to the assumptions and interpretations to improve their clarity and ensure their validity. In combination with adding a section on the study’s limitations, we believe that the Results and Discussion sections are

improved, and we are grateful for the suggestion from the reviewer to undertake these changes.

13. The first sentence of the Discussion doesn't really make sense. Do the authors mean the discovery of rare variants deepens our understanding of prostate cancer?

As part of the rewriting of the Discussion section this sentence has been removed.

14. The first paragraph of the Discussion ends with "Independently, the genetic variants..." Apart from AOX1, whose association was only suggestive, I'm not sure the authors can say the discovered rare variants independently play a role in cancer severity. BRCA2 maybe, but an association of BRCA2 in the non-aggressive cases vs. controls would need to be determined in order to be more certain. Based on the study's findings, I find it hard to accept that ATM plays an independent role.

We thank the reviewer for raising the lack of clarity in this paragraph and have rewritten it so that it better reflects our results:

"Analysing associations between germline variation and disease end-points provides insight into the distinct pathogenic roles of individual genes (<https://doi.org/10.1101/2023.10.10.23296544>). Specifically, we identify germline variants in the case versus control analysis that play a role in the overall risk of developing prostate cancer, while genetic variants identified in the within-case aggressive versus non-aggressive analysis play a role in prostate cancer severity. In our study, *BRCA2* was the only gene with clear evidence for a role both in the overall risk of prostate cancer and also in determining the degree of aggressiveness, consistent with previous reports (PMID: 37733366). In comparison, two other genes – *SAMHD1* and *CHEK2* – showed significant associations in the case-control analysis of prostate cancer risk but demonstrated no association with disease severity, similar to the reported effect of *HOXB13* p.Gly84Glu (PMID: 25595936). Conversely, damaging germline variants in *AOX1* were not associated with the overall risk of developing prostate cancer, but were associated with aggressive disease at the suggestive level. This is consistent with a prior GWAS identifying a common variant at the *AOX1* locus, which was associated with prostate-cancer-specific survival time, and with *AOX1* expression levels that correlated with disease recurrence (PMID: 30289108). If validated, this implies a role for *AOX1* in prostate cancer progression, but no substantial impact on the overall risk of disease development."

15. Second paragraph of the Discussion, "Beyond the DDR, we identify...", should perhaps be changed to "we validate or identify rare non-synonymous variants..." given variants in *ANO7* and *HOXB13* have previously been identified.

In the now rewritten version of the manuscript, this sentence has been edited and focuses on the gene-level results:

“Here, we validate *BRCA2*, *ATM* and *CHEK2* deleterious rare variants as significant risk factors, and reproduce the recently described association of *SAMHD1* (PMID: 39261734) with prostate cancer in UKB and replicate the finding in additional cohorts.”

16. I find the last point of the above paragraph a little confusion. Do the authors mean the utility of pop. genetics is to identify men who may benefit from certain therapies or is the utility to inform therapeutic strategies? If the variant in AR is protective, it could be argued that you wouldn't want to give men ADT, as this would diminish AR levels and potentially protection. However, this isn't a gene that is associated with severe prostate cancer, so it's therapeutic potential isn't great regardless. I feel that these variants are more suited to developing better screening strategies.

We are grateful for the reviewer's comments on this point. Using the *AR* example, we intended to highlight the utility of population genetics to identify potential therapeutic targets. Indeed, there is strong evidence that drug targets with human genetic evidence connecting gene to disease are more likely to be successful in clinical trials (PMID: 26121088). Specifically in this example, we suggest that the association of prostate cancer with germline variants in the androgen receptor pathway, an already established therapeutic target in prostate cancer, serves as proof of the principle that human population genetics can be leveraged for drug target discovery.

Addressing the reviewer's comment about direction of effect, since the predicted damaging missense variant in *AR* reduces risk of prostate cancer (i.e. is protective), we would expect that it would have the same functional effect as therapeutic inhibition of androgen receptor signalling, an approach already in widespread clinical use in prostate cancer management (PMID: 35372068).

We agree with the reviewer that the germline variants highlighted in our study are suited to developing improved strategies for screening and risk stratification, and we highlight this in the Discussion (lines 355-365). In addition, we suggest that they can also reveal potential therapeutic targets, captured in the following re-worded sentence:

“The protective missense variant in *AR*, which encodes the androgen receptor, is notable given the widespread treatment of prostate cancer patients with anti-androgen therapies (PMID: 19796750), and highlights the connection between rare germline variant disease associations and potential therapeutic targets.”

17. The paragraph discussing *SAMDH1* (p.6-7) is also a little confusing/disjointed. It needs to be made clear that the same QV model found an association between *SAMHD1* and telomere length, as it currently reads as though just the model was the same. Shorter telomers are also more often associated with increased cancer risk, not reduced. While I realise the opposite has also been suggested, I'd suggest reading up a little more on the association between telomers and cancer risk (e.g., Okamoto and Seimiya have a review paper) and revising this paragraph.

This paragraph has now been rewritten to improve clarity and we explicitly state that the same QV model is associated with prostate cancer and telomere length (TL) for *SAMHD1*:

“It is notable that the QV model strongly associating *SAMHD1* with prostate cancer risk here is the same model we recently found to be associated with longer telomere length (PMID: 39192095).”

We also thank the reviewer for highlighting the review article by Okamoto and Seimiya (PMID: 30709063). The review article states “it has been reported that the telomere length of prostate cancers is shorter compared with normal tissues”. This is most likely due to the increased rate of cell division in prostate cancer cells, and the review article states that “telomeres in human somatic cells gradually become shortened with each cell division.” In fact, the relationship between TL and somatically driven clonal expansions is complex. For example, in pre-malignant blood clonal expansions, termed clonal haematopoiesis of indeterminate potential (CHIP), Nakao et al conclude that “Bidirectional MR supported the hypotheses that longer leucocyte telomere length (LTL) promotes CHIP acquisition, while CHIP in turn shortens LTL potentially among affected cells” (PMID: 35385311). In a separate study, Codd et al confirm the association of longer genetically determined LTL with higher risk of prostate (among many other) cancers, and also suggest that “longer TL predisposes individuals to an increased risk of cancer (as also supported by the MR analysis), but as tumor cells proliferate, cells within the tumor demonstrate a shorter TL” (PMID: 34611362).

To clarify that we are referring to TL in normal tissues (reflecting background telomere length influencing cancer risk), as opposed to TL specifically within prostate cancer cells, we include the term “leucocyte telomere length”:

“Given the widely reported links between telomere biology and cancer (PMID: 34611362, 27498151, 37140166), in particular the association between longer genetically predicted leucocyte telomere length and increased prostate cancer risk (PMID: 34611362), telomere maintenance is implicated as a potential mechanism for *SAMHD1*-mediated predisposition to prostate cancer.”

18. The last sentence of the Discussion mentions prostate cancer prevention, how has this study provided evidence for therapeutic approaches to prevention? Do the authors mean their work has implications for distinct screening and therapeutic approaches?

It is reasonable to suppose that genes/variants associated with the risk of developing prostate cancer, such as *BRCA2*, *ATM* and *SAMHD1*, play a role in malignant transformation, i.e. the transition from normal to cancer. In contrast, we suggest that genes associated with prostate cancer severity, but not with overall risk of disease, such as *AOX1*, are likely to play a role predominantly in progression of established cancer, rather than mediating the earlier transformation from normal to cancer. This distinction could help to guide therapeutic strategies in future. For example, following the above logic, we would predict that therapeutic modulation of *AOX1* would be futile in future prostate cancer prevention strategies, but might be a useful component in treatment of established disease.

In the extensively rewritten Discussion section, we have removed the specific reference to therapeutic approaches to prevention and, instead, include the statement that “the

identification of pathogenic variants in specific genes/pathways could inform precision medicine strategies.”

19. In the Methods section (second paragraph), is it correct that the MCPS cohort has had no phenotype updates, whether through linkage to a cancer registry or death records since 2004? Or was the linkage to death registry data done more recently? Can this be clarified?

The clinical data was updated through linkage to the death register in 2020. We thank the reviewer for pointing out this omission, and have now added this information to the Methods section (line 389).

Response to Reviewers

Manuscript “Assessing the contribution of rare protein-coding germline variants to prostate cancer risk and severity in 37,184 cases”

Reviewer #2 (Remarks to the Author):

The authors have addressed all of my main concerns, and their changes have not introduced any new issues. The only remaining issues are extremely minor and, I think, can be left to the discretion of the authors.

We are pleased to hear that the main concerns have been addressed in a satisfactory manner. For the remaining very minor points, we have replied inline below.

In the numbering of my initial review:

4. I still think the failure to account for population stratification could be acknowledged more fully as a limitation of the paper. This issue seems to me just as applicable for sequencing studies as GWASs, but more problematic for sequencing studies because it's harder to adjust for ancestry. If a variant is only present in people from Northern Finland, and Northerners are more likely to develop prostate cancer than Southerners due to unrelated lifestyle differences, then wouldn't you say this rare variant is associated with prostate cancer? Though I accept that this issue is similar to the more important issue of cases and controls being drawn from different populations, which you acknowledge.

Specifically for the Finnish cohort, the ExWAS was performed with REGENIE which has been shown to control for Type 1 errors in rare variants (PMID: 34017140). However, we understand the Reviewer's concern and emphasise in the Discussion section that variants found to be associated with prostate cancer in FinnGen require replication in additional cohorts:

“Secondly, in our single variant analysis, while the Finnish population represents a powerful bottleneck population for discovering low frequency disease-associated variants (PMID: 36653562), the extreme rarity of many of these variants in non-Finnish European populations makes replication of findings challenging, even in large cohorts such as UKB.”

8. For the TERT variant, all counts were NA. Should these be zero instead?

For single variant testing, in each cohort we set a minimum minor allele count of 8 for inclusion. In Supplementary Table 8, NA indicates that the minimum minor allele count of 8 was not reached for that cohort.

Reviewer #3 (Remarks to the Author)

Most of the responses to my comments are reassuring, especially the forest plots.

I agree that the unknowns regarding aggressive disease would bias toward the null. Given that this study did not discover many (any?) novel variants, the concern remains. The authors have now acknowledged this limitation, which is adequate.

My remaining concern is regarding covariates. If I understood it correctly, the authors' response explains that Fisher's exact test is computationally efficient and has been useful in other publications/studies. That is all fine, but I am not sure it answers my question. If there is an imbalance between cases and controls in a known confounder (e.g., age, genetic ancestry, or family history), this could yield misleading results for genetic associations.

In addition to its computational efficiency, we have demonstrated that Fisher's exact test (FET) provides a robust assessment of associations for rare variants. In our previous publications (PMIDs 34375979, 37794183, 25700176, and 33326012), we have demonstrated that our pre-association cohort harmonization steps ensure the robustness of the test statistics to false positives. Specifically, we have shown that FET performs comparably to a generalized mixed model approach that includes sex, age, and ancestry principal components as covariates, and superior when studying sparse observations as occurs with rare variant statistics (PMID 34375979, response to Referee 1 Comment 3 here).

However, we acknowledge the reviewers request and to ensure our experiences also apply to the results presented in this manuscript, we have now also conducted Firth logistic regression with age, age², and ancestry principal components included as covariates for the largest cohort (UKB) in our gene-level analyses. We observed highly consistent test statistics between Fisher's exact test and Firth logistic regression with the Fisher's exact tests overall generating more conservative statistics than the Firth regression alternative. We now include these comparable Firth logistic regression statistics in the supplementary tables (Supplementary Table 3 columns AQ-AT & Supplementary Table 7 columns Y-AB, and referenced in Methods "Gene-level Collapsing Analysis"). Combined with the absence of genomic inflation observed (Supplementary Tables 2 and 6), we believe this further evidences the veracity of our results.

Reviewer #4 (Remarks to the Author):

I co-reviewed this manuscript with one of the reviewers who provided the listed reports. This is part of the Nature Communications initiative to facilitate training in peer review and to provide appropriate recognition for Early Career Researchers who co-review

manuscripts.

Thank you for participating in the Nature Communications initiative to support Early Career Researchers. We appreciate the collaborative effort in reviewing our manuscript.

Reviewer #5 (Remarks to the Author):

The Authors have made substantial changes to the manuscript and their description of the results based on the Reviewers' comments. As such, the manuscript is much improved and more clearly reflects the data presented.

Thank you for your positive feedback, and we appreciate the opportunity revising our manuscript gave us to enhance our work.